# Epigenetic modification of the PD-1 (*Pdcd1*) promoter in effector CD4+ T cells tolerized by peptide immunotherapy

Rhoanne C McPherson[1], Joanne E Konkel[1], Catriona T Prendergast[1], John P Thomson[2], Raffaele Ottaviano[2], Melanie D Leech[1], Oliver Kay[1], Stephanie E J Zandee[1], Claire H Sweenie[1], David C Wraith[3], Richard R Meehan[2], Amanda J Drake[4], Stephen M Anderton[1]*

[1]MRC Centre for Inflammation Research, University of Edinburgh, Edinburgh, United Kingdom; [2]MRC Human Genetics Unit, Institute of Genetics and Molecular Medicine, University of Edinburgh, Edinburgh, United Kingdom; [3]Department of Cellular and Molecular Medicine, University of Bristol, Bristol, United Kingdom; [4]University/BHF Centre for Cardiovascular Science, University of Edinburgh, Edinburgh, United Kingdom

**Abstract** Clinically effective antigen-based immunotherapy must silence antigen-experienced effector T cells (Teff) driving ongoing immune pathology. Using CD4+ autoimmune Teff cells, we demonstrate that peptide immunotherapy (PIT) is strictly dependent upon sustained T cell expression of the co-inhibitory molecule PD-1. We found high levels of 5-hydroxymethylcytosine (5hmC) at the PD-1 (*Pdcd1*) promoter of non-tolerant T cells. 5hmC was lost in response to PIT, with DNA hypomethylation of the promoter. We identified dynamic changes in expression of the genes encoding the Ten-Eleven-Translocation (TET) proteins that are associated with the oxidative conversion 5-methylcytosine and 5hmC, during cytosine demethylation. We describe a model whereby promoter demethylation requires the co-incident expression of permissive histone modifications at the *Pdcd1* promoter together with TET availability. This combination was only seen in tolerant Teff cells following PIT, but not in Teff that transiently express PD-1. Epigenetic changes at the *Pdcd1* locus therefore determine the tolerizing potential of TCR-ligation.

*For correspondence: steve.anderton@ed.ac.uk

**Competing interests:** The authors declare that no competing interests exist.

## Introduction

Antigen-based immunotherapy remains the 'holy grail' of immune tolerance because it should target only those pathogenic lymphocytes driving autoimmune, allergic, or alloreactive immunopathology, whilst leaving beneficial immune surveillance unaltered. Peptide immunotherapy (PIT) is the subject of clinical trials in autoimmune and allergic disease (*Larche and Wraith, 2005*; *Larché, 2007*). Despite this the molecular basis for effects of PIT, particularly on T effector (Teff) cells, which is the clinical imperative, remains to be fully understood.

The co-inhibitory receptor PD-1 (encoded by the *Pdcd1* locus) is transiently upregulated on both CD4+ and CD8+ T cells upon activation in order to restrain primary immune responses (*Agata et al., 1996*; *Keir et al., 2006*, *2008*). Its role in maintaining peripheral tolerance under steady state conditions is illustrated by the spontaneous development of autoimmune pathology in mice that lack PD-1 (*Nishimura et al., 1999*). PD-1 is also highly-expressed on exhausted CD8+ T cells (*Barber et al., 2006*; *Youngblood et al., 2013*). PD-1 contains an immunoreceptor tyrosine-switch motif (ITSM) that is thought to recruit SHP-2, a phosphatase that can inhibit the PI3K pathway (*Zhang et al., 2002*; *Chemnitz et al., 2004*). Signalling through PD-1 upon TCR stimulation has been shown to inhibit proliferation and the production of IL-2 and effector cytokines by T cells (*Freeman et al., 2000*;

**eLife digest** The immune system protects the body from dangerous microbes and removes damaged cells. However, in some cases, the immune system can malfunction and attack healthy tissues, which can lead to type-1 diabetes, multiple sclerosis, and other autoimmune diseases. Many of the current treatments for these disorders suppress the immune system, which can make the individuals more susceptible to infections.

It may be possible to treat autoimmune diseases using small pieces of protein—called peptides—that are based on proteins found on the cells that the immune system attacks by mistake. This strategy would target the specific immune cells that are malfunctioning, but allow the rest of the immune system to continue to work as normal. Peptide-based therapies for autoimmune diseases are currently being tested in clinical trials, and although the results look promising, it is not known precisely how they work.

McPherson et al. used mice that develop a disease similar to multiple sclerosis because some of their immune cells, known as effector T cells, attack a protein found in the mouse brain called MBP. The mice were treated with a peptide based on part of MBP, which prevented them from developing the autoimmune disease. The success of the peptide therapy depended on the T cells producing large amounts of a protein called PD-1. This protein stops the T-cells from activating immune responses when they detect the MBP protein.

The gene that makes PD-1 can have a methyl-tag—a chemical modification to DNA—which alters how much PD-1 is made in the T cells. When the gene has this methyl-tag, it can only be switched on for a short time to make a small amount of PD-1, which helps to control the immune responses activated by the T cell. However, when the methyl-tag was removed as a result of the peptide therapy the gene could be switched on for much longer, so that much more PD-1 was produced.

This work helps us to understand how peptide therapy works and should improve the chances of using this therapy to successfully treat patients with autoimmune diseases.

*Sandner et al., 2005*; *Keir et al., 2006*). The importance of PD-1 signalling in PIT has been unclear. Reversal of unresponsiveness has been reported in CD8[+] T cells upon blockade of PD-1 signalling (*Tsushima et al., 2007*; *Chikuma et al., 2009*), but PD-1 was dispensable for both the induction and maintenance of tolerance in PIT-exposed naïve CD4[+] T cells (*Konkel et al., 2010*).

In the clinical setting, PIT is required to control activated Teff cells during ongoing inflammation. Although PIT has been reported to reverse clinical signs of disease (*Leech et al., 2007*), this scenario has been seldom explored mechanistically. An understanding of this is clearly of major importance to successful clinical translation. Here we used a peptide of myelin basic protein (MBP) and MBP-responsive TCR transgenic cells to show that PIT was capable of silencing Teff cells, thereby preventing murine experimental autoimmune encephalomyelitis (EAE). PD-L1[hi] CD4[+] dendritic cells (DC) were uniquely capable of providing sustained presentation of peptide-MHC (pMHC) complexes following PIT. PD-1-deficient T cells were resistant to PIT. In PD-1-sufficient Teff, PIT drove demethylation of the *Pdcd1* promoter, correlating with loss of 5-hydroxymethylation (a potential DNA demethylation intermediate) and lasting PD-1 expression. These data help define an epigenetic signature of T cell tolerance following PIT and therefore have implications for the development of protein biomarkers for clinical efficacy in current and anticipated tolerogenic modalities.

## Results

### Non-deletional tolerance in response to PIT

The Ac1-9(4Tyr) peptide of MBP, containing a Lys→Tyr substitution at residue 4 of the peptide, is a potent tolerogen when administered in soluble form either to wildtype (WT) H-2[u] mice or to Tg4 mice expressing a transgenic TCR responsive to this peptide (*Liu and Wraith, 1995*; *Burkhart et al., 1999*). To trace a defined antigen-responsive cohort of T cells we adapted these protocols by prior transfer of naïve CD4[+] Tg4.CD45.1 T cells into B10.PL (H-2[u]), or B10.PLxC57BL/6 (H-2[u,b]) mice. These F[1] mice are resistant to EAE induced with the MBP peptide, unless first seeded with a cohort of Tg4 T cells (*Ryan et al., 2005*). Tracing the presence and function of the transferred Tg4 cells is therefore of direct

relevance as they are the pathogenic T cell population in these experiments. A single i.v. injection of the MBP peptide protected against subsequent efforts to induce EAE by immunization (*Figure 1A*). Donor T cells persisted in the spleen (*Figure 1B,C*), but there was reduced production of IFN-γ and IL-17, in splenic recall assays amongst PIT-treated mice (*Figure 1C* and *Figure 1—figure supplement 1*). Of note, we found no evidence for an elevation in the frequency of Foxp3+ donor Tg4 cells, nor in IL-10 production in response to PIT (*Figure 1C*). We concluded from these initial studies that a single exposure to the MBP peptide was sufficient for successful PIT, without enhanced induction of cell death, or establishment of Treg-mediated suppression, but rather an intrinsic unresponsiveness in the persisting Tg4 cells.

## Stable pMHC complexes presented by CD4+ DC after PIT

Splenocytes isolated from mice administered Ac1-9(4Tyr) were potent in vitro stimulators of a Tg4 T cell line (Tg4.TCL) and remained so beyond 72 hr after peptide injection (*Figure 2A*). Splenic populations were MACS-sorted into B cell (CD11c−CD19+) and DC (CD11c+) populations. Whilst DC were more efficient presenters on a per cell basis than unfractionated splenocytes, B cells were very poor presenters (*Figure 2A*). Although the B cell populations used contained fewer than 2% of CD11c+ cells, it is plausible that this level of contamination accounted for the Tg4.TCL stimulation evident with high numbers of B cells isolated at the 2-hr time-point. The ability of DC to present the peptide-MHC complex in vivo was underlined by immunofluorescent staining showing transferred Tg4 cells in areas of the spleen rich in CD11c+ cells (*Figure 2—figure supplement 1*).

Splenic CD11c+ cells from MBP peptide-treated mice were sorted into CD11cintPDCA-1+ (pDC) CD11chiCD4+CD8− (CD4+ DC), CD11chiCD4−CD8+ DC (CD8+ DC) and CD11chiCD4−CD8− populations (*Figure 2B*). The ability to stimulate the Tg4.TCL was only maintained in the CD4+ DC (*Figure 2C*). Isolation of splenic macrophage populations from peptide-treated mice provided no evidence that these cells could maintain the pMHC complex (data not shown).

## PIT requires T cell expression of PD-1

The CD4+ DC population is located in the T zone of the spleen and so is ideally placed to present tolerogenic peptide to naive T cells (*McLellan et al., 2002*). CD4+ DC from steady state mice had particularly high expression of PD-L1 in comparison to other DC populations (*Figure 2D,E*). Transferred Tg4 T cells expressed high levels of PD-1 in response to PIT, in contrast to those from PBS-treated mice (*Figure 3A,B*). Addition of an anti-PD-1 blocking antibody to ex vivo rechallenge cultures restored the ability of splenocytes from PIT-treated mice to produce IFN-γ and IL-17 (*Figure 3C*).

Collectively, the above data indicated that sustained pMHC presentation during PIT can drive high expression of PD-1 in the responding T cells. This, coupled with pMHC presentation specifically by PD-L1hi CD4+ DC renders the T cells unable to produce pro-inflammatory cytokines. To determine the functional importance of PD-1 in this tolerogenic process, we generated Tg4.PD-1−/− mice. Transferred naïve Tg4 cells from these mice were insensitive to PIT, going on to cause a typical course of EAE upon subsequent immunization that was indistinguishable from that induced with Tg4.PD-1+/+ cells that had not been subjected to PIT (*Figure 3D*). We therefore conclude that PD-1 plays a non-redundant role in this model of CD4+ T cell tolerance.

## PIT abrogates Teff cell function and CNS infiltration

Although PIT can inhibit ongoing EAE (*Leech et al., 2007*), the mechanisms behind this have been less well explored than for the effects of PIT on naïve T cells. We therefore determined the effects of PIT using a previously established passive EAE model in which Teff cells are generated from naïve Tg4 T cells prior to transfer into WT hosts (*O'Connor et al., 2010*). PIT rendered the host mice profoundly resistant to disease (*Figure 4A*). As with naïve T cells shown above, the Teff cells persisted after PIT-treatment, but importantly these cells failed to populate the CNS efficiently (*Figure 4B*). Instead, they were present in the spleen in sufficient numbers to allow detailed analysis following retrieval by FACS-sorting (*Figure 4C*). PIT did not drive Foxp3-expression in Tg4 Teff cells (*Figure 4D*), nor was IL-10 production evident (not shown). Rather, within 4 days of PIT, proinflammatory cytokine production by the transferred Teff cells was profoundly diminished (*Figure 4E,F*). The loss of IFN-γ producing capacity in PIT-exposed Teff was not reflected in any loss in T-bet expression (*Figure 4—figure supplement 1*).

The poor accumulation of Tg4 Teff cells within the CNS after PIT (*Figure 4B*) suggested that their ability to home to the target organ might also have been affected. Although the molecular requirements for T cell entry into the CNS during EAE are complex and controversial (*Prendergast and Anderton, 2009*), others have described a requirement for P-selectin glycoprotein ligand-1 (PSGL-1)

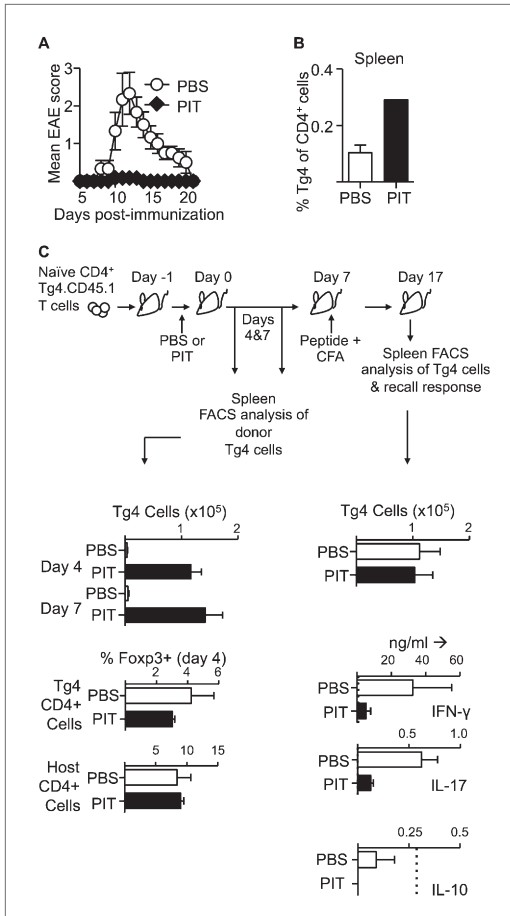

**Figure 1**. PIT induces unresponsiveness in naïve Tg4 cells. (**A**, **B**) B10.PLxC57BL/6 mice received PBS or PIT i.v. 1 day after transfer of naïve CD4+ Tg4 cells. EAE was induced 7 days later by immunization with Ac1-9. (**A**) Mean clinical scores ± SEM. (**B**) Frequency of CD4+ Tg4 cells in the spleen at day 19 post-EAE induction (six mice per group, from one of three experiments giving consistent results). (**C**) Spleens were sampled four and 7 days after PIT/PBS for analysis of CD4+ Tg4 numbers and Foxp3 expression in host and donor CD4+ T cells (3–4 mice per group, from one of three experiments giving consistent results). A separate cohort were immunized on day 7 after PIT/PBS and spleens analyzed 10 days later for CD4+ Tg4 cell numbers and the production of IFN-γ, IL-17 and IL-10 by splenocytes in response to stimulation with 100 μM Ac1-9 (dotted lines represent cytokine levels for unstimulated controls) (four mice per group, from one of three experiments giving consistent results).

The following figure supplement is available for figure 1:

**Figure supplement 1**. PIT reduces the frequency and number of pro-inflammatory cytokine producing Tg4 cells.

in the pathogenic process (*Deshpande et al., 2006*). We have also found that encephalitogenic function of Tg4 Teff populations correlates with their elevated expression of PSGL-1 and that an antibody to PSGL-1 can inhibit Tg4 Teff-driven EAE (*Figure 4G*), presumably by interfering with the interaction of PSGL-1 on the Teff cells with P-selectin and/or E-selectin on the CNS endothelium. Whilst transferred Tg4 Teff cells exposed to PBS showed elevated PSGL-1 expression, their counterparts that had been exposed to PIT showed PSGL-1 expression more comparable with that seen on the EAE-irrelevant CD4+ cells of the host mice (*Figure 4H*), PSGL-1 glycosylation mediated by 2 β-1,6-N-acetyl glucosaminyl-transferase and α(1,3)-fucosyltransferase-VII) is required for functionality (binding to selectins) (*Yu et al., 2000*; *Smithson et al., 2001*; *Sperandio et al., 2001*; *Deshpande et al., 2006*). Staining with a mouse P-selectin-human IgG fusion protein indicated that active PSGL-1 was also reduced on Tg4 Teff cells in response to PIT (*Figure 4—figure supplement 2*). These data suggest that one consequence of PIT is disrupted PSGL-1-dependent T cell trafficking.

## Sustained PD-1 expression is required for Teff cells to be silenced by PIT

We also observed that PD-1 expression was high on Teff cells retrieved from PIT-treated mice compared to PBS-treated controls and that this difference was maintained beyond 2 weeks from peptide administration (*Figure 5A–C*). This was not a consequence of any long-term pMHC presentation in the lymphoid system, because PIT-treated Teff cells that had been retrieved from their first hosts maintained PD-1 (and did not induce EAE) when transferred into secondary hosts that were not exposed to PIT (*Figure 5D–F*). The tolerant phenotype was therefore stable in T cells exposed to PIT.

As was the case with the earlier experiments using naïve T cells (*Figure 3D*), Tg4.PD-1−/− Teff cells were not tolerized and were able to cause disease in the presence of PIT (*Figure 5G*), also establishing PD-1 as a required component of tolerance in this more therapeutic setting.

## PD-1 limits Teff cell clonal expansion in response to PIT

To understand how the absence of PD-1 signaling might allow the maintenance of pathogenic function in spite of PIT administration, we further compared the function of Tg4.PD-1+/+ versus Tg4. PD-1−/− Teff cells. There were no gross differences

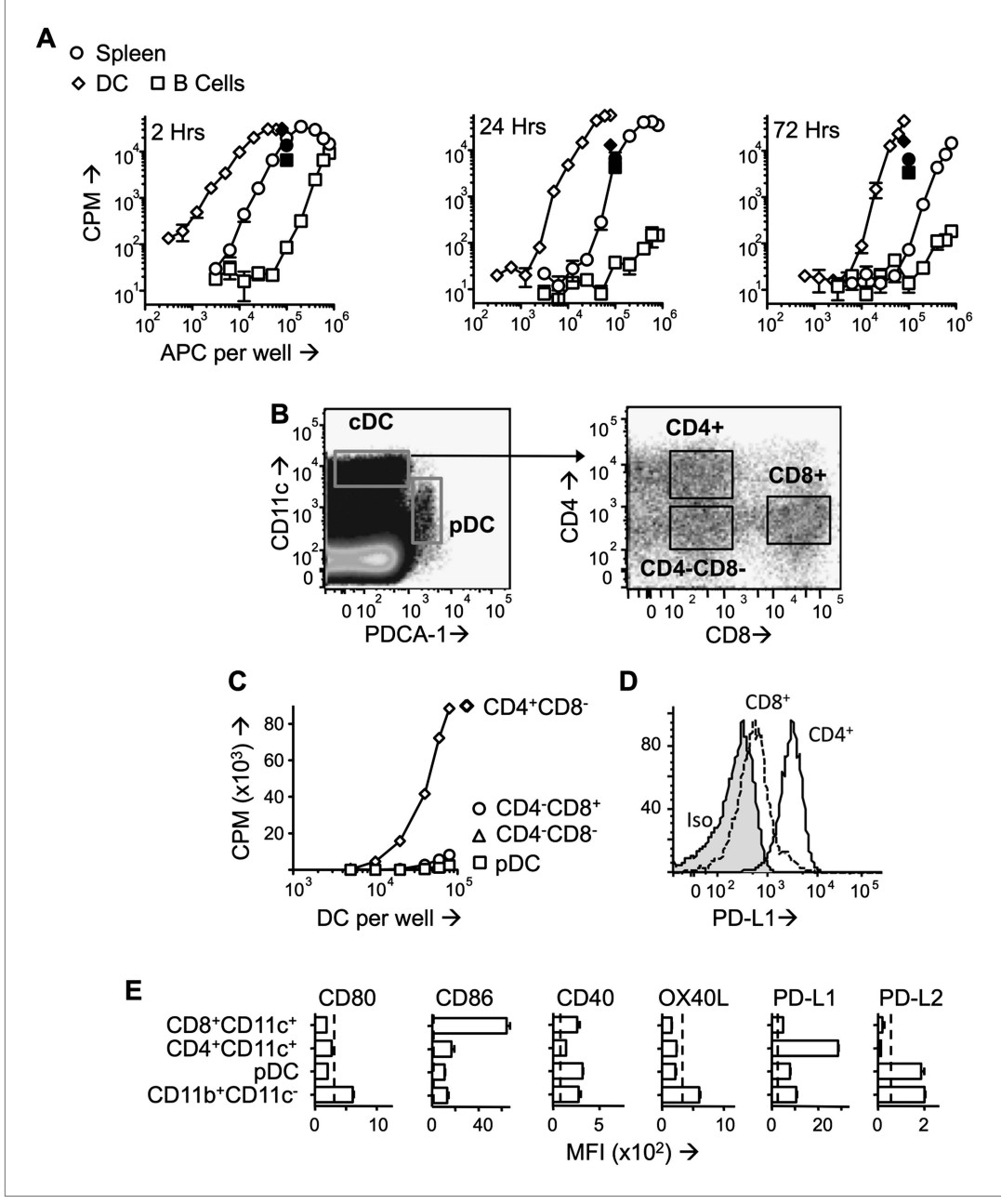

**Figure 2**. PD -L1^hiCD4^+ DC present tolerogenic pMHC complexes to T cells. (**A**) B10.PL mice received PIT i.v. and splenocytes were isolated at the indicated times. B cells (CD11c^–CD19^+) and DC (CD11c^+) were MACS-sorted and their ability to stimulate a Tg4.TCL was measured compared to whole splenocytes. Positive controls (solid symbols) show proliferative responses of Tg4.TCL to DC, B cells and splenocytes in the presence of 20 µM Ac1-9. Data are from one of three experiments giving consistent results. (**B**) Gating strategy for FACS-sorting of DC sub-sets from spleen isolated 1 day after PIT administration. (**C**) Proliferative responses of Tg4.TCL to increasing numbers of DC. Data are from one of three experiments giving consistent results. (**D**) Representative histograms of steady state expression of PD-L1, gated on CD11c^hiCD4^+ and CD11c^hiCD8^+ splenic DC. (**E**) MFI ± SEM of staining with mAb to co-stimulatory and co-inhibitory molecules on steady state splenic APC populations (four mice per group,, from one of two experiments giving consistent results, dotted lines represent MFI of isotype control staining).

The following figure supplement is available for figure 2:

**Figure supplement 1**. Colocalization of transferred naïve Tg4 cells with CD11c+ splenocytes.

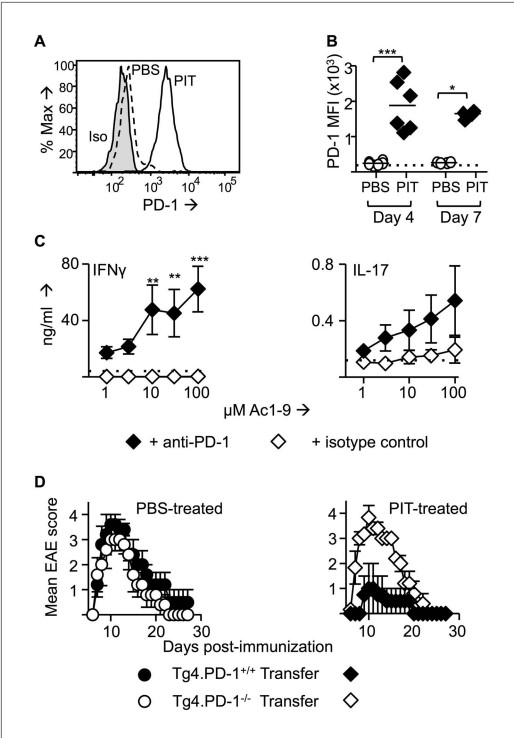

**Figure 3**. PD-1 is required for the establishment and maintenance of tolerance in naïve CD4⁺ Tg4 cells. (**A**–**C**) B10.PL mice received PBS/PIT 1 day after transfer of naïve CD4⁺ Tg4 cells. (**A**) Representative histograms of PD-1 expression gated on CD4⁺ Tg4 donor cells in spleen 4 days after PBS/PIT. (**B**) MFI of PD-1 staining gated on CD4⁺ Tg4 donor cells in spleen 4 and 7 days after PBS/PIT (4–6 mice per group, from one of three experiments giving consistent results, dotted line represents MFI of isotype control staining). (**C**) IFN-γ and IL-17 production in response to Ac1-9 by splenocytes isolated 7 days after PIT and cultured in the presence of anti-PD-1 or isotype (four mice per group, from one of three experiments giving consistent results, dotted lines represent cytokine levels for unstimulated cultures). (**D**) EAE in B10.PLxC57BL/6 mice that received PBS/PIT 1 day after transfer of naïve CD4⁺ cells from Tg4.PD-1⁺/⁺ or Tg4.PD-1⁻/⁻ donors. EAE was induced 7 days after PBS/PIT by immunization with Ac1-9 (five mice per group, from one of three experiments giving consistent results).

in the numbers or frequencies of donor cells in PBS-treated hosts (*Figure 5—figure supplement 1*), indicating that PD-1 expression does not constrain the accumulation of donor cells in the absence of PIT. However, the PIT-driven elevation in donor cell numbers seen with Tg4.PD-1⁺/⁺ Teff cells was accentuated in Tg4.PD-1⁻/⁻ Teff cells and a high frequency of these cells showed expression of the cell cycle marker Ki-67 (*Figure 5—figure supplement 1*). A consequence of this overall increase in the number of Tg4.PD-1⁻/⁻ Teff cells was increased numbers of Foxp3⁺ cells as well as cytokine⁺ cells in this group. However, as seen with Tg4.PD-1⁺/⁺ Teff cells, the frequencies of Foxp3⁺ or cytokine⁺ Tg4.PD-1⁻/⁻ Teff cells were reduced following PIT (*Figure 5—figure supplements 2,3*). In contrast to their reduced effector cytokine production, PIT did not block the ability of Tg4.PD-1⁻/⁻ Teff cells to produce IL-2 upon ex vivo re-challenge (*Figure 5—figure supplement 4*). Tg4 Teff cells retrieved from PBS-treated mice had low expression of CD25 and this was absent in their PIT-treated counterparts (*Figure 5—figure supplement 5A*). After 12 hr culture with MBP peptide, CD25 expression was increased in PBS-exposed Teff cells, irrespective of their PD-1-status, but there was a notable impairment in this upregulation of CD25 in PIT-treated Tg4.PD-1⁺/⁺ Teff cells. This impairment was not evident in PIT-exposed Tg4.PD-1⁻/⁻ Teff cells (*Figure 5—figure supplement 5B*). The deficit in CD25-upregulation seen in PIT-treated Tg4.PD-1⁺/⁺ Teff cells was reflected by a low frequency of cells with phosphorylated STAT5 in response to IL-2. Notably however, PIT-treated Tg4.PD-1⁻/⁻ Teff cells did not show such abrogated STAT5 phosphorylation (*Figure 5—figure supplement 6*). We conclude that the ability of PD-1-deficient Teff cells to drive EAE despite administration of PIT is associated with their heightened capacity for clonal expansion.

## PIT-drives PD-1 expression in non-transgenic Teff cells

To assess whether PIT-driven PD-1 expression was evident in a non-transgenic system, we studied ovalbumin (OVA)-responsive Teff in mice on the C57BL/6 background. Lymph nodes from CD45.1 mice that had been immunized with the OVA(323–339) peptide (pOVA) in CFA provided a source of Teff, which were then transferred into CD45.2 hosts prior to administration of pOVA or PBS (*Figure 5—figure supplement 7A,B*). Reduced frequencies of cytokine⁺ donor cells following in vitro restimulation were indicative of PIT-induced unresponsiveness (*Figure 5—figure supplement 7C,D*). Ex vivo analysis showed clear PD-1ʰⁱ CD4⁺ populations in donor cells from the PIT-treated group (*Figure 5—figure supplement 8A,B*). Focusing on cells within the PD-1⁺ gate, PIT-exposed donor cells showed higher levels of PD-1 expression than PBS-exposed donor cells (*Figure 5—figure supplement 8C*). We conclude that elevated expression of PD-1 associates with abrogation of effector function in response to PIT in this non-transgenic system.

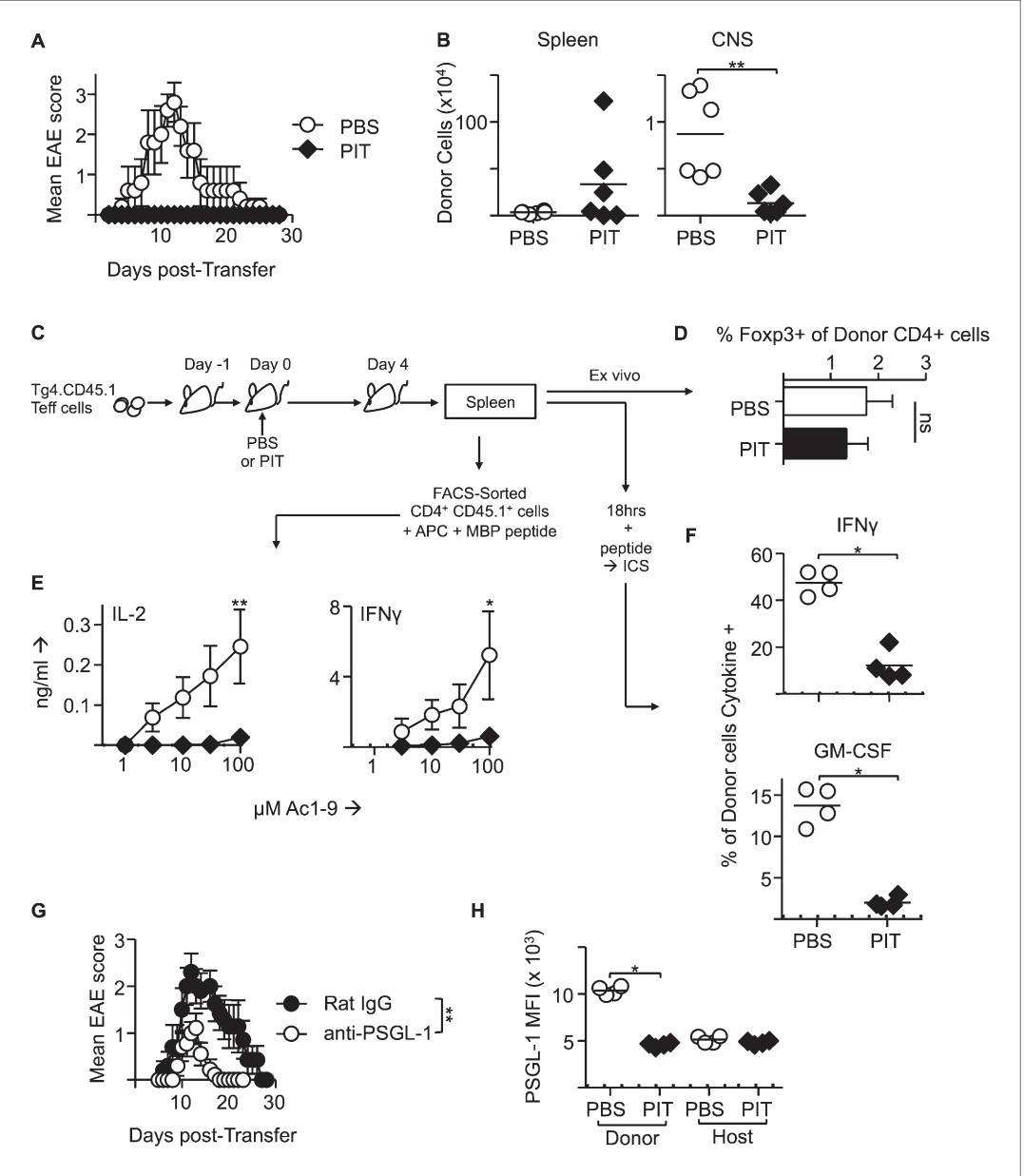

**Figure 4**. PIT induces tolerance in pathogenic Tg4 Teff cells. (**A**) EAE in B10.PL mice that received PBS/PIT 1 day after transfer of Tg4 Teff cells (six mice per group, from one of five experiments giving consistent results). (**B**) Numbers of CD4+ Tg4 donor cells in spleen and CNS of PBS/PIT-treated mice sampled at the peak of EAE (six mice per group, from one of two experiments giving consistent results). (**C**) Tg4 Teff cells were transferred and spleens were sampled 4 days after PBS/PIT. (**D**) The frequencies of Foxp3+ cells amongst the donor Tg4 CD4+ Teff cells on day 4 after PBS or PIT. (**E**) Ac1-9-induced IL-2 and IFN-γ production by retrieved CD4+ Tg4 donor cells (FACS-sorted 4 days after PBS/PIT and cultured with irradiated B10.PLxC57BL/6 splenic APC) (four mice per group, from one of two experiments giving consistent results, dotted lines represent cytokine levels for unstimulated controls). (**F**) GM-CSF and IFN-γ production following overnight culture of splenocytes with Ac1-9 (gated on CD4+ Tg4 donor cells) (n = 4 per group, from one of four experiments giving consistent results, dotted lines represent cytokine levels for unstimulated controls). (**G**) Modulation of EAE by anti-PSGL-1 (50 μg given i.v. on alternate days from day 1 after Tg4 Teff cells). Teff cells were also incubated with 20 μg/ml of the relevant antibody for one hour prior to transfer. (n = 10 per group, from one of three experiments giving consistent results). (**H**) PSGL-1 expression by CD4+ host and Tg4 donor cells 4 days after PBS/PIT (four mice per group, from one of two experiments giving consistent results, dotted line represents MFI of isotype control staining).

*Figure 4. Continued on next page*

*Figure 4. Continued*

The following figure supplements are available for figure 4:

**Figure supplement 1**. Pathogenic Tg4 Teff cells maintain T-bet after PIT.

**Figure supplement 2**. PIT reduces active PSGL-1 levels on Tg4 Teff cells.

## PIT drives demethylation of the *Pdcd1* promoter

PD-1 is rapidly upregulated by T cells upon initial TCR stimulation but is then lost over time (*Riley, 2009*). This is exemplified by the fact that our pathogenic Tg4 Teff cells expressed PD-1 on the day of transfer, but this was lost within 3 days of transfer in PBS-treated hosts (*Figure 5A,C*). In contrast, PIT drove the long-term expression of PD-1 by potentially pathogenic CD4+ T cells and this was a required component of PIT. Sustained expression of PD-1 is important in another scenario in which T cells develop antigen-unresponsiveness; CD8+ T cell 'exhaustion' in the face of chronic viral infection (*Barber et al., 2006*; *Wherry et al., 2007*). So, are similar mechanisms, leading to long-term expression of PD-1, at the heart of these two states of T cell unresponsiveness?

Stable gene expression is often associated with a variety of epigenetic changes within promoter regions, including alterations in the levels of active histone post-translational modifications (PTMs) such as trimethylation of H3K4 (H3K4me3), or repressive histone PTMs such as H3K27me3. Indeed, CD8+ Teff cells with low PD-1 expression were reported to have elevated levels of the repressive H3K27me3 PTM at the *Pdcd1* promoter, compared to either PD-1− naive cells or PD-1hi CD8+ cells from chronically infected mice (*Youngblood et al., 2011*). We measured active (H3K4me3) and repressive (H3K27me3) histone marks at the *Pdcd1* promoter by ChIP-PCR and did not identify similar changes that could distinguish pathogenic Tg4 Teff, from PIT-tolerized Teff. In naïve Tg4 cells and PBS-treated Tg4 Teff cells (neither of which expressed PD-1), neither modification (H3K27me3 or H3K4me3) was enriched at the two conserved regions, CR-B and CR-C (*Youngblood et al., 2011*), of the *Pdcd1* promoter (*Figure 6A,B*). In contrast, both the PD-1 expressing groups (Teff on the day of transfer and PIT-treated Teff) exhibited elevated levels of H3K4me3, but no enrichment for H3K27me3 (*Figure 6A,B*). Importantly, functional (pathogenic) PBS-treated Teff that lacked PD-1 expression did not show elevated levels of the repressive H3K27me3 PTM, thereby distinguishing their behaviour from that reported for PD-1lo CD8+ Teff cells (*Youngblood et al., 2011*). Moreover, it was not possible to distinguish PD-1hi pathogenic Teff (on the day of transfer) from PD-1hi PIT-tolerized Teff based on their relative abundance of H3K4me3 versus H3K27me3 at the *Pdcd1* promoter.

Stable PD-1 expression in exhausted CD8+ T cells is associated with DNA demethylation within the CR-B and CR-C regions of the *Pdcd1* promoter (*Youngblood et al., 2011*). We retrieved Tg4 cells at various stages of the Teff PIT model for DNA methylation analysis. DNA modification was quantified, after bisulfite conversion, using pyrosequencing to assess individual CpG sites within the whole cell population. Cloning and sequencing after bisulfite conversion provided complementary information on the pattern of DNA methylation at CpG sites across individual cloned alleles (*Figure 6C,D*). Consistent with previous analyses of naive CD8+ T cells (*Youngblood et al., 2011*), CpGs within the CR-B region were largely modified (~80% methylation across most CpG sites) in naïve CD4+ Tg4 cells (*Figure 6C–D*), whereas DNA methylation levels were considerably lower across the CR-C region. Despite expression of PD-1 by Tg4 Teff cells on the day of transfer (*Figure 5A*), there were no differences in DNA methylation at the *Pdcd1* promoter of these pathogenic cells compared to naïve Tg4 cells (*Figure 6C,D*), indicating that demethylation is not a pre-requisite for transient PD-1 expression following initial CD4+ T cell activation. As would be expected based on this result, DNA methylation was maintained at CR-B and CR-C in the Teff cells retrieved from PBS-treated hosts, where PD-1 expression had returned to levels observed in naïve T cells (*Figure 5A*). In marked contrast, PIT-exposed Teff, which had prolonged PD-1 expression (*Figure 5A–F*), showed complete demethylation at CR-C and significantly reduced DNA methylation at the CR-B region in the *Pdcd1* promoter (*Figure 6C,D*). Therefore only PIT was capable of altering DNA methylation patterns at the *Pdcd1* promoter. This enables PD-1hi tolerant cells to be distinguished from fully functional Teff, which only transiently express PD-1.

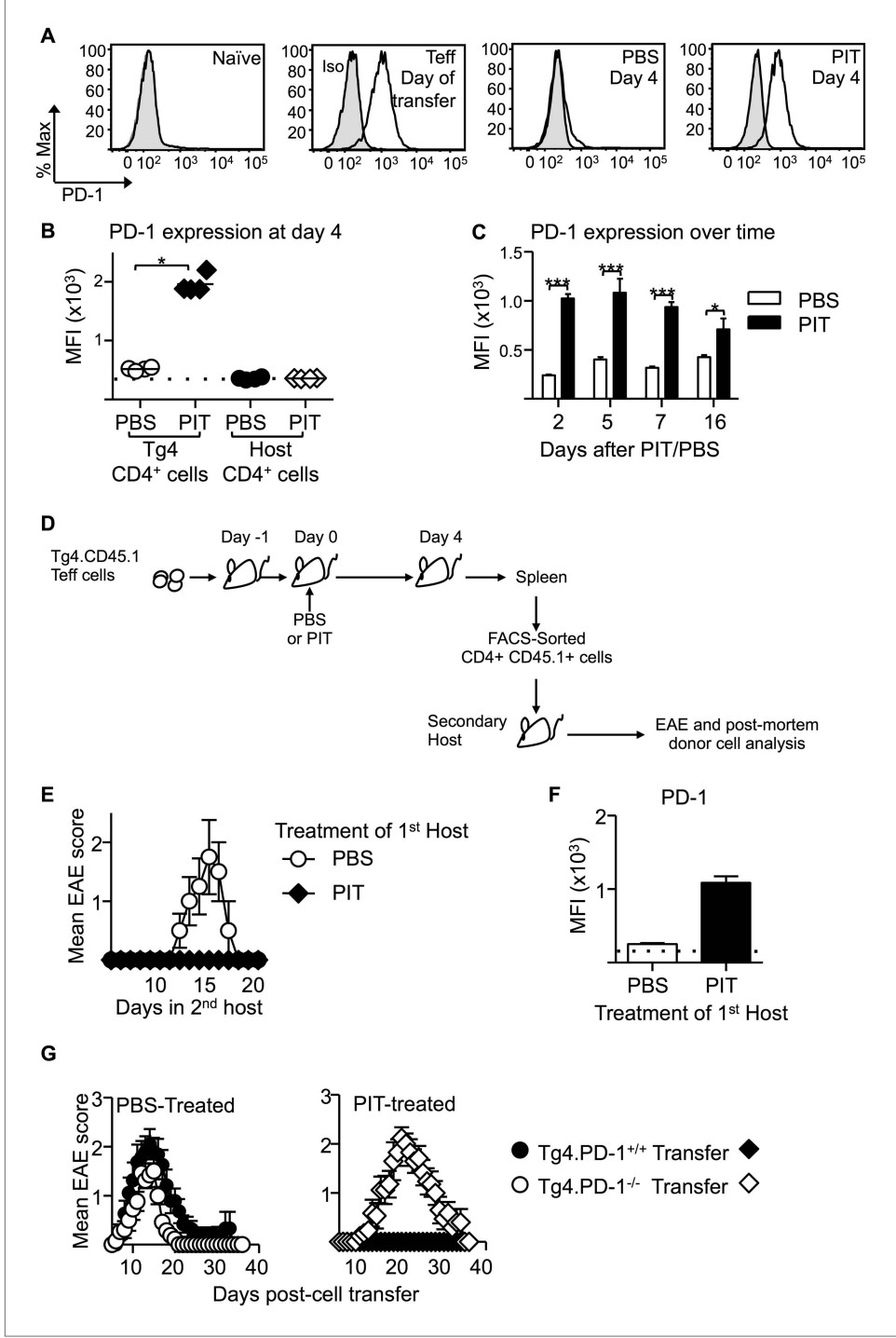

**Figure 5**. PD-1 expression is maintained by PIT and is required for tolerance. (**A**) Representative histograms of PD-1 expression by CD4+ Tg4 cells sampled as naïve cells, Teff on day of transfer and Teff cells retrieved 4 days after PBS/PIT. (**B**) PD-1 expression (MFI) gated on CD4+ host cells and Tg4 donor cells from spleen 4 days after treatment (four mice per group, from one of four experiments giving consistent results, dotted line represents MFI of isotype control staining). (**C**) Time course of PD-1 expression on CD4+ Tg4 donor cells from PBS/PIT-treated mice (four mice per group). (**D**) B10.PLxC57BL/6 mice received PBS/PIT 1 day after transfer of Tg4 Teff cells. 4 days later CD4+ Tg4 donor cells were FACS-sorted and 2 × 10⁶ were transferred into secondary hosts that were not exposed to PIT (PTX was given on the same day). (**E**) EAE in secondary hosts (n = 4 for PBS; 21 for PIT, pooled from two experiments). (**F**) PD-1 expression gated on CD4+ Tg4 donor cells from spleens isolated 16 days after secondary
*Figure 5. Continued on next page*

*Figure 5. Continued*

transfer (dotted line represents MFI of isotype control staining). (**G**) EAE in B10.PLxC57BL/6 mice that received PBS/PIT 1 day after transfer of Teff generated from Tg4.PD-1$^{+/+}$ or Tg4.PD-1$^{-/-}$ donors (n = 20–36 mice per group, pooled from three experiments).

The following figure supplements are available for figure 5:

**Figure supplement 1**. Elevated accumulation of Tg4.PD-1$^{-/-}$ Teff cells following PIT.

**Figure supplement 2**. PIT does not increase the frequency of Foxp3$^+$ Tg4 Teff cells.

**Figure supplement 3**. PIT limits the frequency of cytokine$^+$ Tg4 Teff cells independently of PD-1.

**Figure supplement 4**. PIT-exposed Tg4.PD-1$^{-/-}$ Teff maintain their ability to produce IL-2.

**Figure supplement 5**. PD-1 limits CD25 up-regulation following recall stimulation of PIT-exposed Tg4 Teff.

**Figure supplement 6**. PD-1 limits phosphorylation of STAT5 in Tg4 Teff cells following PIT.

**Figure supplement 7**. PIT limits effector cytokine production by polyclonal Teff cells.

**Figure supplement 8**. PIT drives PD-1 expression in polyclonal Teff cells.

## The *Pdcd1* promoter of naïve CD4$^+$ cells is enriched in 5-hydroxymethylcytosine, which is lost following PIT

The precise processes that lead to demethylation of 5-methylcytosine (5mC) are not fully understood, but direct conversion of 5mC to cytosine seems unlikely. Rather, the favoured model is that 5mC is progressively oxidized to generate intermediates, facilitated by the Ten-Eleven-Translocation (TET) proteins, and these intermediates are subsequently converted either by DNA repair or replication dilution to cytosine (*Kohli and Zhang, 2013*). The first intermediate is 5-hydroxymethylcytosine (5hmC), a stable epigenetic mark with widespread tissue distribution (*Kriaucionis and Heintz, 2009*; *Tahiliani et al., 2009*; *Song et al., 2011*; *Nestor et al., 2012*; *Thomson et al., 2012*). Reciprocal changes for both 5mC and 5hmC at gene loci imply that demethylation may be occurring via a 5hmC intermediate (*Thomson et al., 2013*). Of note, standard bisulfite conversion techniques cannot distinguish 5mC from 5hmC.

Using an affinity based technique (*Thomson et al., 2012*), we identified considerable enrichment of 5hmC at the *Pdcd1* promoter CR-B and CR-C regions in naïve Tg4 cells (*Figure 6E*). 5hmC was also present in Teff cells generated by 3-day culture (particularly within the CR-B region) and in Teff cells retrieved from PBS-treated mice. In contrast, 5hmC was undetectable at CR-B and CR-C in cells retrieved from PIT-treated mice (*Figure 6E*).

We tested whether changes in the relative expression of 5hmC at the *Pdcd1* promoter were reflected by changes in the global expression of TET genes (as assessed by qPCR). All three known mammalian TETs were expressed in naïve CD4$^+$ Tg4 cells, but these levels were substantially reduced in Teff cells (*Figure 6F*). Expression of each TET was partially restored in Tg4 Teff cells retrieved from either PIT or PBS-treated hosts, suggesting these that cells were equipped for the oxidation of both 5mC and 5hmC to occur.

Collectively our data indicate that stable expression of PD-1 is coincident with demethylation of the *Pdcd1* promoter and that this requires a combination of permissive histone changes and oxidation of 5mC and 5hmC involving the activity of TETs. In the system used here, these two requirements only coincide following TCR engagement in response to PIT.

## Discussion

Our data provide key insights into the mechanisms of Teff cell unresponsiveness induced by PIT, which are pertinent to its clinical translation. A role for PD-1 in the establishment, or maintenance, of T cell unresponsiveness has been reported in various experimental models of autoimmune and allergic

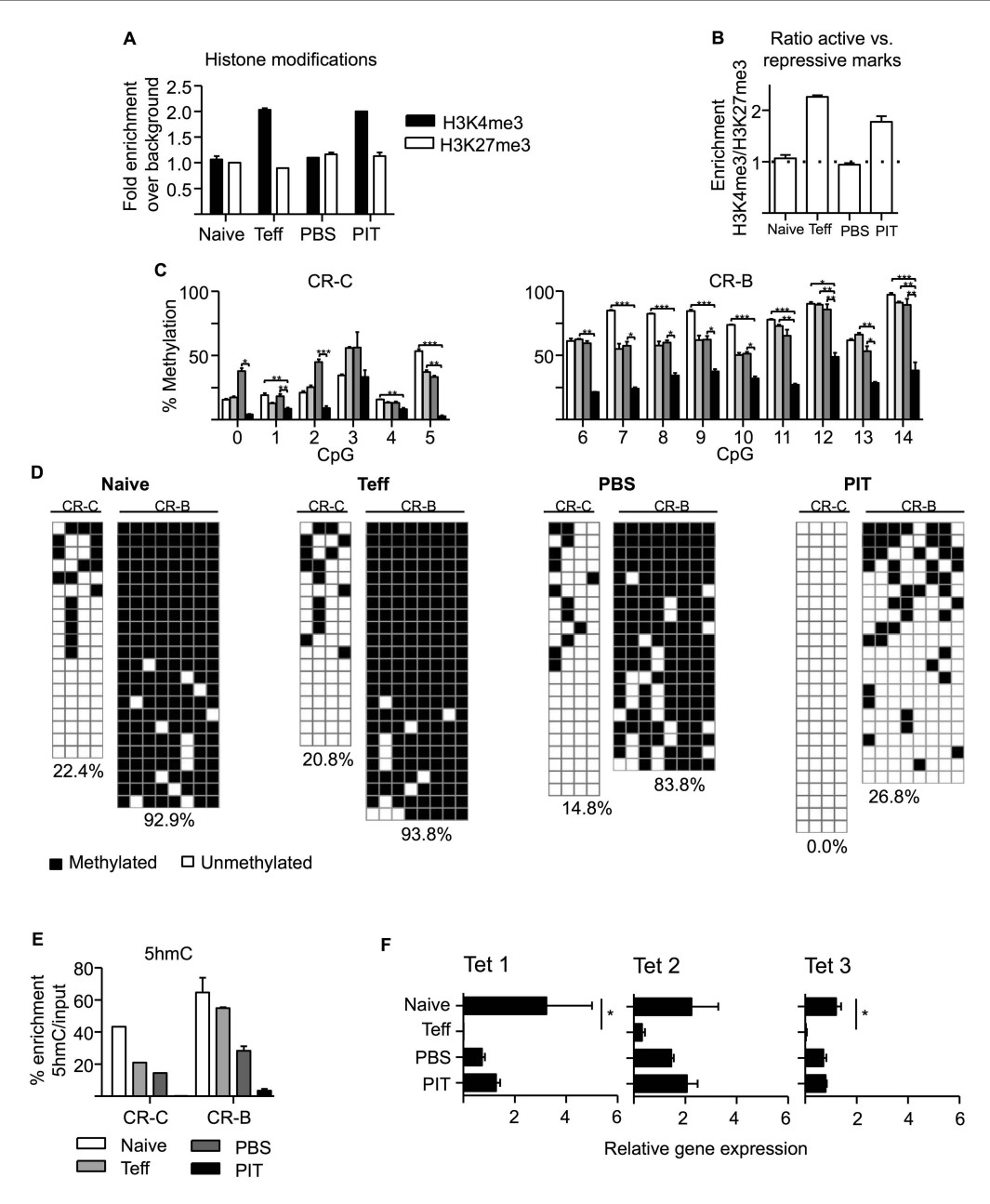

**Figure 6**. PIT induces epigenetic modification of the *Pdcd1* promoter. CD4[+] Tg4 T cells were isolated as naïve cells, Teff on day of transfer and cells retrieved 4 days after either PBS or PIT for analysis of the CR-C and CR-B regions. (**A**, **B**) Histone modifications (H3K4me3 and H3K27me3) were analysed at CR-C and CR-B by ChIP and qPCR. Data in (**B**) show the ratio of H3K4me3 over H3K27me3 fold-enrichment over background and are representative of two independent experiments. (**C**, **D**) DNA methylation status determined by pyrosequencing (**C**) and bisulfite sequencing (**D**) (data are from two (**C**) or one (**D**) bisulfite conversions, four or more mice per group). (**D**) Methylated (black) and unmethylated (white) CpGs. Vertical columns represent individual CpGs (1–4 for CR-C and 7–14 for CR-B). Horizontal rows represent individual cloned alleles. (**E**) 5hmC enrichment at CR-C and CR-B as measured by 5hmC DNA Immunoprecipitation and qPCR. Data are from two experiments using pooled samples. (**F**) Total RNA was extracted from the isolated cell populations and expression levels of TET1, TET2 and TET3 were measured by qPCR. Values are expressed relative to housekeeping gene (GAPDH) and are representative of three experiments.

inflammation (*Probst et al., 2005*; *Cheng et al., 2007*; *Francisco et al., 2010*). Our previous work had excluded such a role for PD-1 in PIT-driven tolerance for naive CD4⁺ T cells, because PIT remained effective irrespective of antibody blockade or genetic deletion of PD-1 in the peptide-responsive T cells (*Konkel et al., 2010*). In the two experimental models used in that study, PIT appeared to drive the abortive activation and apoptosis of peptide-responsive naïve T cells within 4 days, which would override any potential for PD-1-mediated long-term unresponsiveness. In contrast to those previous studies using naïve T cells, we found here that Teff persisted following therapeutic application of PIT. In a recent study using PIT to silence allergic airways inflammation, we found an analogous accumulation of allergic Teff cells, indicating that persistence following PIT is common to Teff with different inflammatory functions (*Mackenzie et al., 2014*).

The observations that pMHC complexes were specifically maintained on CD4⁺ DC in vivo, and that these DC showed the highest constitutive expression of PD-L1, led us to assess the role of PD-1 in PIT here. A recent study transgenically expressed autoantigen on all DC subsets and found PD-1-dependent elevation in the frequencies of autoreactive Foxp3⁺ T cells as a result (*Yogev et al., 2012*). That result may well have reflected an influence of CD8⁺ DC, which can drive de novo Foxp3– expression due to their production of TGF-β (*Yamazaki et al., 2008*). CD4⁺ DC do not have that capacity, accounting for the absence of increased Foxp3-expression here, in Tg4 cells exposed to PIT. This, together with the lack of evidence for IL-10 production in response to PIT suggests that, in this study, the predominant effect of PIT is to suppress pathogenic effector function rather than to drive gain of regulatory function.

PD-1 has been reported to co-locate to TCR microclusters at the immunological synapse, where, following Lck-mediated phosphorylation of its ITSM, PD-1 recruits the SHP-2 phosphatase to counteract phosphorylation of TCR-proximal signalling machinery (*Yokosuka et al., 2012*). Use of Tg4.PD-1⁻/⁻ Teff cells indicated a key role for PD-1 in limiting clonal expansion following PIT. There are several ways in which this inhibition could be achieved. Inhibition of PI3K/Akt and MEK/ERK pathways can lower cell proliferation by reducing transcription of SKP2. This gene encodes a crucial component of the ubiquitin ligase SCF^Skp2, which is required for cyclin-dependent kinase 2 activation and cell cycle progression from G1 into S phase (*Parry et al., 2005*; *Appleman et al., 2006*; *Patsoukis et al., 2012*; *Pauken et al., 2013*). Exogenous IL-2 only partially restores cell proliferation by activating the MEK/ERK pathway but not the PI3K/Akt pathway (*Patsoukis et al., 2012*), suggesting that IL-2 cannot reverse PD-1-dependent unresponsiveness. Recruitment SHP-2 to the cytoplasmic tail of PD-1 is thought to function primarily by inhibiting the PI3K pathway (*Chemnitz et al., 2004*). However, SHP-2 has also been implicated in the dephosphorylation and inactivation of pSTAT5 (*Yu et al., 2000*; *Chen et al., 2003*) and PD-1 signalling has been associated with a decrease in both the expression and phosphorylation of STAT5 (*Iliopoulos et al., 2011*). These observations correlate with the decreased levels of pSTAT5 we see in Tg4 Teff cells (with intact PD-1 signalling) following PIT. Furthermore, inactivation of STAT5 can lead to a reduction in cell-surface CD25 expression (*Kim et al., 2001*). This, coupled with the inhibition of TCR signalling pathways by PD-1, would likely account for the impaired CD25 expression that we saw in PIT-exposed Tg4 Teff cells. An impaired ability to upregulate CD25, and therefore to respond to IL-2, is distinct from the classical profile for clonal anergy, in which unresponsive T cells can upregulate CD25 but cannot produce their own IL-2 (anergy is broken by addition of exogenous IL-2) (*Schwartz, 2003*). An unresponsive state that blocks CD25 upregulation can be viewed as more secure, since it removes the risk of IL-2 from a bystander source overcoming that unresponsiveness.

Consistent with their lack of encephalitogenicity, Teff that had been tolerized by PIT had an impaired ability to establish themselves in the CNS. This might reflect their impaired ability to undergo clonal expansion upon subsequent exposure to MBP in the CNS, as discussed above. A recent report has indicated an additional role for PD-1 in inhibiting autoimmune T cell infiltration into the pancreas in diabetes (*Pauken et al., 2013*). Here we observed that tolerized Teff had lower levels of active PSGL-1 relative to pathogenic T cells (PBS group), which is an important component of pathogenic T cell entry into the CNS in this model. To our knowledge, no direct causal relationship between PD-1 and PSGL-1 has been established. T follicular helper (TFH) cells are PD-1^hi and PSGL-1^lo , but we do not believe that PIT drives Teff towards a TFH fate, since we did not observe any production of the TFH-associated cytokine IL-21, nor did we find evidence for any gain of the TFH-associated transcription factor BCL-6 following PIT (data not shown). The transcription factor T-bet can inhibit PD-1 expression in CD8⁺ cells (*Kao et al., 2011*), but here we found that this was not the case for CD4⁺ Teff cells, which were T-bet⁺ PD-1^hi following PIT, indicating that T-bet and PD-1 are not necessarily mutually antagonistic.

Our data allow us to develop a model to explain the pronounced demethylation of the *Pdcd1* promoter that, in the experimental system used here, is a unique characteristic of Tg4 Teff that have been tolerized in response to PIT. The *Pdcd1* promoter of Tg4 cells in their naïve state appears to be fully methylated, although this includes a significant level of 5-hydroxymethylation. This, together with the availability of TETs 1–3 and the absence of the repressive H3K27me3 histone mark, suggests that the gene is poised for transcriptional activation and PD-1 expression. This would fit with the known early expression of PD-1 following initial TCR ligation (as seen here in Teff on the day of transfer). This PD-1 expression during Teff generation was mirrored by chromatin changes associated with gene transcription (H3K4me3 enrichment), but without a contemporaneous demethylation of the *Pdcd1* promoter. The considerable reduction in TET expression observed at the Teff stage should impede any active demethylation through the oxidation of either 5mC or 5hmC (a process that involves TET activity), thereby averting long-term PD-1 expression that would affect Teff function. As these cells transition to a PD-1-negative Teff state (pathogenic Tg4 cells retrieved from PBS-treated hosts), histone modifications associated with active transcription are no longer evident and, although TET expression has been restored, this repressive chromatin state prevents the TET (and other) proteins from accessing the *Pdcd1* promoter to allow demethylation. However, a key feature of the tolerogenic properties of PIT is to provide long-term PD-1 expression by tolerant CD4$^+$ T cells, which is associated with the presence of stable histone modifications characteristic of active transcription (thus enabling access of the TET proteins, amongst others), and therefore loss of both 5mC and 5hmC at the *Pdcd1* promoter. These data indicate that analysis of the methylation/hydroxymethylation status of the *Pdcd1* promoters may facilitate the discrimination of tolerant versus recently activated PD-1$^+$ CD4$^+$ T cells.

Juvenile arthritis patients who responded to PIT using peptide from the DNAj antigen were reported to have an 'immune signature' that included elevated global PD-1 expression, relative to non-responders (*Koffeman et al., 2009*). However, that work was based on assessment of the patients before undergoing PIT, rather than following PIT. It would therefore be important to understand the relative expression of PD-1 before and after treatment in individual patients in future studies. A limitation is the need to identify T cells that are relevant to the peptide(s) used for PIT. As the range of pMHC tetramers available expands, this should become more feasible. Furthermore, our study highlights the need for analysis of the *Pdcd1* promoter to discriminate PD-1$^{hi}$ pathogenic Teff from the PD-1$^{hi}$ Teff rendered tolerant by PIT. This would be challenging to combine with tetramer-based isolation of relatively small T cell populations, although perhaps not impossible. Therefore, whilst locus-specific DNA methylation-status would not prove a viable biomarker in itself, it might aid the identification of a surrogate phenotypic profile amenable to flow cytometric analysis.

The demethylation of the *Pdcd1* promoter seen in PIT-tolerized Teff is reminiscent of the changes reported to occur in CD8$^+$ T cells exhausted in response to chronic viral infection (*Youngblood et al., 2011*). So, do these two situations of CD4$^+$ T cell tolerance and CD8$^+$ T cell exhaustion have the same underlying mechanism(s)? In some ways yes, but there are some pronounced differences. In both scenarios Teff cells express PD-1 but this is lost, either as they transition to memory cells (*Youngblood et al., 2011*), or here within 2 days of transfer in the absence of PIT. Conversely, exhausted or peptide-tolerized Teff cells maintain PD-1 expression. Youngblood et al reported that CD8$^+$ Teff cells that transiently expressed PD-1 already have demethylated *Pdcd1* promoters and their transition towards PD-1-negative memory cells is driven by the specific remethylation of their *Pdcd1* promoters (*Youngblood et al., 2011*). In contrast, the methylation status of the *Pdcd1* promoters within our CD4$^+$ Teff cells that transiently expressed PD-1 appeared unaltered from that of naïve CD4$^+$ T cells. It was only in the PIT-treated Teff that we saw significant demethylation of the *Pdcd1* promoter, consistent with their sustained PD-1 expression. It will be important to determine whether these differences reflect intrinsic differences in how PD-1 expression is controlled in CD4$^+$ versus CD8$^+$ T cells. In addition, we are not aware of any information on 5hmC status of the *Pdcd1* promoter in CD8$^+$ cells.

We found marked enrichment of 5hmC at the *Pdcd1* promoter in naïve CD4$^+$ T cells. This is intriguing because the biological significance of 5hmC has chiefly been explored in the context of embryonic development. Its appearance in the promoter of a key gene controlling T cell activation suggests that DNA hydroxymethylation may play a broader role in maintaining the 'pluripotent' state of naïve CD4$^+$ T cells. We suggest that this warrants global analyses to search for 5hmC and other intermediates of 5mC oxidation within the CD4$^+$ T cell genome and also for the TET proteins with direct roles in the oxidation process (*Deplus et al., 2013*; *Koh and Rao, 2013*).

## Materials and methods

### Mice, antigens and tissue culture medium

C57BL/6 (CD45.1 or CD45.2), B10.PL (CD45.2), B10.PLxC57BL/6 (CD45.2) and Tg4 (CD45.1 or CD90.1) mice (*Liu et al., 1995*) were used. PD-1$^{-/-}$ mice (kindly provided by Prof Tasuku Honjo, Kyoto University) were crossed with Tg4 (CD90.1) mice to obtain Tg4.PD-1$^{-/-}$ (CD90.1) mice. All mice were bred under specific pathogen-free conditions at the University of Edinburgh. All experiments were approved by the University of Edinburgh Ethical Review Committee and were performed in accordance with UK legislation. Acetylated MBP peptides Ac1-9 (Ac-ASQKRPSQR) and Ac1-9 (4Tyr) (Ac-ASQYRPSQR) and the ovalbumin peptide 323–339 (pOVA) (ISQAVHAAHAEINEAGR) were synthesised with C-terminal amides by Cambridge Research Biochemicals, Billingham, UK. Tissue culture medium (RPMI 1640 medium) was supplemented with 2 mM L-glutamine, 100 U/ml penicillin, 100 µg/ml streptomycin, and $5 \times 10^{-5}$ M 2-ME (all from Invitrogen Life Technologies, Paisley, UK) and 10% FCS (Sigma, Poole, UK). Cells isolated from immunized mice were cultured in X-VIVO15 serum-free medium (Lonza, Walkersville, MD, USA) supplemented with 2 mM L-glutamine and $5 \times 10^{-5}$ M 2-ME.

### T cell transfers and retrieval

Naïve CD4$^+$ T cells were isolated from Tg4 or Tg4.PD1$^{-/-}$ mice by magnetic activated cell-sorting (MACS) according to manufacturer's instructions (Miltenyi Biotec, Teterow, Germany), and $2 \times 10^6$ cells were transferred in 200 µl PBS i.v. Pathogenic MBP-responsive Teff cells were generated by stimulation of Tg4 or Tg4.PD1$^{-/-}$ splenocytes with Ac1-9 in the presence of IL-12, IL-18 and IL-2 as described previously (*O'Connor et al., 2010*). pOVA-responsive Teff cells were generated by immunization of C57BL/6(CD45.1) mice with pOVA, followed by restimulation of draining lymph node cells with pOVA and the above cytokine cocktail. Cells were harvested after 72 hr culture and $2 \times 10^6$ blasts transferred i.v. in 200 µl PBS. For the retrieval of donor Tg4 cells, CD4$^+$ T cells were isolated from splenocytes by MACS and donor cells (CD4$^+$ CD45.1$^+$ or CD90.1$^+$) were identified and isolated by surface staining and FACS using a FACSAria II (Becton Dickinson, Franklin Lakes, NJ, USA).

### Immunizations and administration of tolerogenic peptide

On the day indicated, mice received Ac1-9 or pOVA emulsified in CFA containing 50 µg heat-killed *Mycobacterium tuberculosis* H37Ra (Sigma) in a total volume of 100 µl injected subcutaneously (s.c.) into the hind legs. For PIT, mice received 200 µg of Ac1-9 (4Tyr), or pOVA, in 200 µl PBS (or PBS alone) i.v. at the indicated time.

### Induction of EAE

For the induction of active EAE, mice were immunized as above and received 200 ng of pertussis toxin (Health Protection Agency, Dorset, U.K.) in 0.5 ml PBS i.p on the day of immunization and 48 hr later. Clinical signs of EAE were assessed using the following scoring index: 0, no signs; 1, flaccid tail; 2, impaired righting reflex and/or gait; 3, partial hind limb paralysis; 4, total hind limb paralysis; 5, hind limb paralysis with partial front limb paralysis; 6, moribund or dead. For the induction of passive EAE, mice received $2 \times 10^6$ Tg4 effector cells via i.v. injection. On the same day, or the day following cell transfer, mice received 200 ng pertussis toxin in 0.5 ml PBS i.p. and clinical signs were assessed as described above. For some experiments, an anti-PSGL-1 antibody (4RA10, rat IgG1, BioXCell, West Lebanon, NH, USA), or purified rat IgG (Sigma) were administered, as indicated.

### Analysis of splenic APC function after PIT

Mice received the Ac1-9(4Tyr) peptide as described above for the induction of PIT. Spleens were isolated at the indicated times, injected with 100 µl of 8 mg/ml of collagenase IV (Worthington Biochemical Corp, Lakewood, NJ, USA) and incubated at 37°C for 20 min before manual disaggregation. In some experiments, APC populations were purified by positive selection using MACS and anti-CD11c beads (DC), or by negative selection with anti-CD11c beads, followed by positive selection using anti-CD19 beads. For isolation of DC populations, CD3$^+$ and CD19$^+$ cells were depleted by MACS prior to staining and FACS for pDC (PDCA-1$^+$CD11c$^{int}$), CD4$^+$DC (CD11c$^{hi}$PDCA-1$^-$CD4$^+$CD8$^-$), CD8$^+$DC (CD11c$^{hi}$PDCA-1$^-$CD8$^+$CD4$^-$), and CD4$^-$CD8$^-$DC (CD11c$^{hi}$PDCA-1$^-$CD8$^-$CD4$^-$). APC populations were used to stimulate a Tg4 T cell line (Tg4.TCL; generated as previously described [*Anderton et al., 1998*]), with or without the addition of exogenous Ac1-9 peptide to the cultures. Cells ($2 \times 10^4$ Tg4.TCL with a range of APC numbers per well) were cultured in flat-bottom 96-well microtitre plates

(Becton Dickinson) for 72 hr [³H]thymidine incorporation over the final 16 hr was measured using a liquid scintillation β-counter (LKB Wallac, Turku, Finland).

## Assesment of lymphoid recall assays

Splenocyte suspensions were cultured in 96-well flat-bottom microtitre plates (Becton Dickinson) at $8 \times 10^5$ spleenocytes/well. For purified donor T cells, $2 \times 10^4$ T cells were cultured with $2 \times 10^5$ irradiated (30 Gy) splenic APC's/well. Cultures were stimulated with a dose range of Ac1-9 and supernatants were tested for production of IFN-γ and IL-17 at 72 hr by ELISA. Where stated, neutralizing anti-PD-1 antibody (RMP1-14, rat IgG2a), or rat IgG (Sigma) were added to cultures at 10 µg/ml. The RMP1-14 antibody was a gift from Dr Hideo Yagata (Juntendo University, Japan).

## Antibodies and FACS analysis

Cells were stained using the following Abs and isotype controls (all from eBioscience, Hatfield, UK, except where stated); anti-CD4-AF700 (Invitrogen), anti-CD4-eFluor450, anti-CD8-(APC/PE), anti-CD45.1-(FITC/PE), anti-CD90.1-FITC, anti-CD11b-APC, anti-CD11c-PE-Cy7, anti-PDCA-1-eFluor450, anti-PD-1-PE, anti-CD162-PE, anti-CD80 biotin, anti-CD86 PE, anti-PD-L1 APC, anti-PD-L2 APC, anti-OX-40L biotin, andti-CD40 PE, anti-IFN-γ-APC, anti-GM-CSF-PE (Becton Dickinson), anti-Ki-67-PerCP-eFluor710, anti-Foxp3-eFlour450, Streptavidin APC, rat IgG1-(FITC/APC/PerCPCy5.5), rat IgG2a-PE, Rat IgG2a-PerCP-eFluor710, rat IgG2b-(PE/biotin), Armenian hamster IgG-PE. Active PSGL-1 levels were detected by staining cells with a P-Selectin-human IgG fusion protein (BD) and anti-human IgG-PE. Flow cytometric data were acquired using a BD LSRFortessa cell analyzer (Becton Dickinson) and data analysed using FlowJo software (Treestar version 3.2.1, Ashland, OR, USA). For intracellular staining in response to peptide, cells were re-suspended at $8 \times 10^6$/ml in the presence or absence of 20 µM Ac1-9, or pOVA, as appropriate. After overnight culture, 1 µl/ml of brefeldin A (eBioscience, 1000× stock) was added for the last four hours of culture. Cells were washed once in FACS buffer (PBS, 2% FCS, 0.01% $NaN_3$) and surface stained prior to processing for intracellular staining using proprietary buffers according to the manufacturer's instructions (e-bioscience for transcription factor staining or Becton Dickinson for cytokine staining). After incubation in fix/perm buffers, cells were stained for intracellular antigens. Detection of pSTAT5 was performed as previously described (*Leech et al., 2013*) following 12 hr culture in the presence of 20 µM Ac1-9.

## Immunofluorescence

$4 \times 10^6$ Tg4 cells were transferred to mice 1 day after PBS/PIT. Spleens were isolated 48 hr later and embedded and frozen in OCT (Cellpath, Newtown, UK). 6 µm thick sections were fixed in ice-cold acetone. Non-specific binding and endogenous biotin were blocked with 3% BSA (Sigma) and an avidin-biotin blocking kit (Vector Laboratories, Peterborough, UK) respectively. Tissue sections were stained with hamster anti-CD11c-biotin (Biolegend, San Diego, CA, USA), rat anti-CD4-FITC and mouse anti-CD45.1-APC (eBioscience). Biotinylated antibodies were detected with streptavidin-Alexa Fluor 555 (Invitrogen). Sections were mounted in Permafluor mounting medium (Thermo Scientific, Hemel Hempstead, UK). To determine specific binding, secondary antibodies alone or isotype controls (IgG-Biotin [Biolegend], IgG2a-FITC, IgG2a-APC [eBioscience]) were used. Images were acquired with a Leica confocal laser scanning microscope SP5 and LAS AF software (Leica, Wetzlar, Germany) and processed with ImageJ/Fiji software (NIH, Bethesda, MD, USA).

## DNA Methylation analysis

Genomic DNA was isolated from frozen cell pellets of CD4⁺ T cells using the DNeasy blood and tissue kit (Qiagen, Crawley, UK) according to manufacturer's instructions, and ≤1 µg of DNA was subjected to bisulfite conversion using the Epi Tect Bisulfite Kit (Qiagen). Primers for Pyrosequencing (Eurofins MWG Operon, Ebersberg, Germany) were designed for conserved regions C and B (CR-C and CR-B) (*Table 1*) within the *Pdcd1* (PD-1 encoding) gene using PyroMark Assay Design 2.0 software (Qiagen). Pyrosequencing was performed using PyroMark reagents with PyroMark Q24 instrumentation, and data were analysed using PyroMark Q24 software 2.0 (all Qiagen). Background non-conversion levels were between 2–3%. For bisulfite sequencing, genomic DNA was subjected to bisulphite conversion using the EZ DNA Methylation Gold Kit (Zymo Research, Irvine, CA, USA). Bisulfite sequencing was conducted as previously described (*Hackett et al., 2012*; *Reddington et al., 2013*). Primers specified in *Table 2* were used to amplify specific regions of interest with Platinum *Taq* (Invitrogen). A single band was excised, gel extracted, and cloned into pGEM-T-easy (Promega, Madison, WI, USA) for sequencing.

**Table 1.** Primer sequences for pyrosequencing analysis

| PDCD1 region | Assay | CpG | PCR primers 5′-3′ | Sequencing primer 5′-3′ | Product length (bp) |
|---|---|---|---|---|---|
| CR-C | 1 | 1 and 2 | F: AGGTATAAAGGAGGTTTTGTAATAGT | GAGGTTTTGTAATAGTTAGG | 186 |
|  |  |  | R: CCTCAACCACCCAAATTCAAA-BIO |  |  |
|  | 2 | 3, 4, 5 | F: TGGGTGGTTGAGGTAGTT | GTTGAGGTAGTTGTTAGAT | 253 |
|  |  |  | R: CACCTCACCTCCTACTTATCTCT-BIO |  |  |
|  | 3 | 6 | F: TGTTTATTTTAGGGTGGTGAGATTTAT | GTTAGGTATTATGTATGTATATAAG | 221 |
|  |  |  | R: TAAAAACCCACCTCACCTCCTACTT-BIO |  |  |
| CR-B | 1 | 8, 9, 10, 11 | F: AAAGGAAGAAAAGTTTTAAGAGAAAGTAAG | CTATCCCACATACTCC | 167 |
|  |  |  | R: ACCCAACTATCCCACATACT-BIO |  |  |
|  | 2 | 12, 13, 14, 15 | F: GGGTTTTTGTTTTTTAGTAGTAAAGGATTA | ATTAAGGTATAGTTTAGGGTA | 164 |
|  |  |  | R: AAAACCAAACTCTTATCCCTTTAAAA-BIO |  |  |
|  | 3 | 16 and 17 | F: GGTAGGGGAGGGTTTAGT | GTTTTGGGAGTTAAGG | 193 |
|  |  |  | R: TCCTCTCCATTTCTAACCCCTCTTATA-BIO |  |  |
|  | 4 | 7 | F: AGGGTAGTAGAGTTAGTAAATTTAAGATA | AGTAGAGAAAATAGTGAGAT | 103 |
|  |  |  | R: CTCTTAAAACTTTTCTTCCTTTCAAAATAC-BIO |  |  |

## 5hmC DNA Immunoprecipitation

Genomic DNA was extracted as above, and 1 µg of DNA was sonicated (Bioruptor, Diagenode, Liege, Belgium) to produce fragments 200–500 bp (mean 300 bp) in length. Enrichment of 5hmC DNA was carried out using the Active Motif Hydroxymethyl Collector kit (Active Motif, La Hulpe, Belgium) according to manufacturer's instructions.

## Chromatin immunoprecipitation for histone modification analysis

Cross-linked ChIP was conducted essentially as previously described (*Hackett et al., 2012*) using $1–3 \times 10^6$ cells per IP. In brief; after fixation in 1% formaldehyde, cells were lysed and chromatin was sonicated (Bioruptor, Diagenode) to produce fragment sizes 200–1000 bp in length. Chromatin was pre-cleared by incubation with Protein A magnetic beads (Invitrogen), before incubation with antibody bound beads. Antibodies used were anti-H3K27me3, anti-H3K4me3 and the isotype control rabbit IgG (Merck Millipore, Billerica, MA, USA and Santa Cruz Biotechnology, Dallas, TX, USA, respectively). Beads were washed, DNA was eluted and cross-linking was reversed before DNA was purified using the QIAquick PCR purification kit according to manufacturer's instructions (Qiagen). Eluted DNA was quantified using real-time PCR.

## Quantitative real-time PCR (qPCR) analysis

qPCR was conducted on DNA obtained after ChIP or 5hmC enrichment using SYBR Green Real-time PCR master mix (Roche Diagnostics, Burgess Hill, UK) and the primers listed in *Table 2*. Reactions were performed on the Roche LightCycler 480. For ChIP analysis, fold enrichment of specific enrichment over background was calculated using values normalized to input. For 5hmC analysis, relative enrichment was calculated to a known amount of input DNA and all data normalised to a negative control

**Table 2.** Primer sequences for bisulfite sequencing and RT-PCR

| Region | Assay | CpG | PCR primers 5'-3' | Product length (bp) |
|---|---|---|---|---|
| CR-C* | | 0–4 | F: ATAAAGGTATAAAGGAGGTTTTG | 363 |
| | | | R: CCTAACTAACTAATCATTTCT | |
| CR-C (nested)* | | 1–4 | F: GAGGGGATTTTTTTAGTTTTTGT | 289 |
| | | | R: AATTTAATCATTCTACTCTCT | |
| CR-B* | | 7–14 | F: GAAAGGAAGAAAAGTTTTAAG | 332 |
| | | | R: AAACTAAAACCAAACTCTTATCC | |
| CR-B (nested)* | | 7–14 | F: GTTTTTTTGAATTTATAGGGGTG | 276 |
| | | | R: ACTCTTATCCCTTTAAAAAAT | |
| CR-C† | 1 | 3–5 | F: AGTTGCCAGATGGTTTCCAG | 154 |
| | | | R: CTGGGGCATTCTGATGATTT | |
| CR-B† | 1 | 10–14 | F: CGGGTCCTAGGAAATGTTCA | 236 |
| | | | R: GCCAGACTCTTGTCCCTTTG | |
| | 2 | 7–10 | F: TACAGGGGTGTCTGGAGAGG | 156 |
| | | | R: ATGCCCTGAGCTATGCCTTA | |
| GAPDH† | | | F: CCACTCCCCTTCCCAGTT | 147 |
| | | | R: CCTATAAATACGGACTGC | |

*Primer sequences for bisulfite sequencing.
†Primer sequences for RT-PCR.

region at the gapdh promoter. Levels of 5hmC were tested over one region at the CR-C and two at the CR-B which were combined to generate an average value of 5hmC enrichment.

For detection of Tet1, Tet2 and Tet3 mRNA, total RNA was isolated using the RNeasy kit and cDNA was generated using the QuantiTect Reverse Transcription kit (both Qiagen) according to manufacturer's instructions. Quantification of mRNA was conducted by qPCR using pre-designed TaqMan assays from Applied Biosystems (Life technologies, Catalogue no's: Mm01169087_m1, Mm00524395_m1, Mm00805756_m1) as previously described (*Drake et al., 2010*). Results were normalized to the expression of the housekeeping gene GAPDH (Applied Biosystems gene expression assay Mm99999915_g1).

## Statistics
Statistical analysis of results was performed using the Mann–Whitney U test or the Kruskal–Wallis with Dunn's post-hoc test (p values: *<0.05, **<0.01, ***<0.005).

## Acknowledgements
Work in SMA's lab was supported by grants G0801924 and G1100084 from the Medical Research Council (MRC), the Wellcome Trust (087833) and the UK Multiple Sclerosis Society (899/08). Work in RRM's lab was supported by the MRC, the BBSRC and by the Innovative Medicine Initiative Joint Undertaking (IMI JU) under grant agreement number 115001 (MARCAR project). AJD was supported by the MRC and the Scottish Funding Council. The authors have no conflicting interests to declare.

## Additional information

### Funding

| Funder | Grant reference number | Author |
|---|---|---|
| Medical Research Council | G0801924, G1100084 | Rhoanne C McPherson, Melanie D Leech, Stephen M Anderton |
| Wellcome Trust | 087833 | Stephen M Anderton |

| Funder | Grant reference number | Author |
|---|---|---|
| Multiple Sclerosis Society | 899/08 | Rhoanne C McPherson, Stephen M Anderton |

The funders had no role in study design, data collection and interpretation, or the decision to submit the work for publication.

## Author contributions

RCM, JEK, SMA, Conception and design, Acquisition of data, Analysis and interpretation of data, Drafting or revising the article; CTP, JPT, RO, MDL, OK, SEJZ, CHS, Acquisition of data, Analysis and interpretation of data; DCW, Drafting or revising the article, Contributed unpublished essential data or reagents; RRM, Conception and design, Drafting or revising the article; AJD, Conception and design, Analysis and interpretation of data, Drafting or revising the article

## Ethics

Animal experimentation: This study was approved by the University of Edinburgh Ethical Review Panel and was performed in accordance with UK legislation (PPL 60/4116).

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
