## [Decision Letter]

[Editors’ note: this article was originally rejected after discussions between the reviewers, but the authors were invited to resubmit after an appeal against the decision.]

Thank you for choosing to send your work entitled “Epigenetic modification of the PD-1 (*Pdcd1*) promoter in effector CD4^+^ T cells tolerized by peptide immunotherapy” for consideration at *eLife*. Your full submission has been evaluated by Tadatsugu Taniguchi (Senior editor) and 2 peer reviewers, one of whom is a member of our Board of Reviewing Editors, and the decision was reached after discussions between the reviewers. We regret to inform you that your work will not be considered further for publication.

As shown below, both reviewers have expressed concerns about lack of mechanistic insights into how PD-1-high anergic T cells develop and whether the findings with TCR transgenic T cells can be extended to tetramer+ non-transgenic T cells. Epigenetic modification of the PD-1 promoter in PIT-tolerized CD4^+^ T cells has some novelty but more in-depth study is required.

*Reviewer #1*:

This report shows that peptide immunotherapy (PIT) of TCR-transgenic mice induces PD-1-expressing anergic T cells from naïve or effector CD4^+^ T cells and that DNA demethylation of the promoter region of the PD-1 gene is characteristic of such anergic T cells. This sustained PD-1 high expression accompanying epigenetic change in transgenic CD4^+^ T cells after PIT is interesting. Other findings in Figures 1, 2, 3 and 4 are not so novel.

Specific comments:

1) In Figure 5, it is not clear how PIT induces high expression of PD-1 in CD4^+^ T cells. Is it primarily attributed to high expression of PD-L1 by DCs presenting peptide/MHC? If so, how does PIT induce high PD-L1 in DCs? Can peptide/MHC-expressing PD-L1+ DCs induce anergy in transgenic CD4^+^ T cells in vitro?

Does transfer of such peptide-pulsed PD-L1-high DCs induce PD-1-high anergic T cells in vivo?

2) Can these findings with transgenic T cells be extended to non-transgenic mice? For example, does PIT induce MBP-tetramer+ PD-1-high anergic T cells in normal mice?

*Reviewer #2*:

McPherson and colleagues provide new insights into mechanisms of T cell unresponsiveness induced by peptide immunotherapy (PIT). Using a peptide of myelin basic protein (MBP) and MBP specific TCR transgenic T cells, they develop a model of non-deletional tolerance to PIT and demonstrate that PD-1 is required for PIT induced unresponsiveness, as PD-1 deficient T cells are resistant to PIT. In further studies they show that PIT sustains PD-1 expression in naïve or effector T cells. Epigenetic studies reveal that PIT leads to demethylation of the *Pdcd1* promoter in tolerized T cells. These findings are of fundamental and translational importance. However, several aspects of the work need to be clarified, and several of the conclusions seem premature from the data presented, as discussed below:

1) Studies in Figure 1 show that administration of MBP peptide i.v. leads to persistence of Ag-specific donor T cells. Data show that IFN-g and IL-17 production was reduced in splenic recall assays. Is this due to a difference in cytokine production on a per cell basis or the frequency of the cytokine producing cells? How do the percentages (and numbers) of IFN-g and IL-17 producing cells in PIT-treated and control mice compare? Also is there a difference in the frequency of IL-10 producing cells? Is T cell survival affected by PIT in this model?

2) Studies in Figure 3 show that PIT treatment of recipients of naïve Tg4.PD-1^−/−^ T cells do not protect from EAE in contrast to PIT treatment of recipients of Tg4.PD-1^+/+^ T cells. This is a key finding that merits deeper analyses. How do Tg4T cell numbers, as well as percentages of FoxP3^+^, IFN-g and IL-17+ Tg4T cells, in the spleen and CNS compare in the PIT treated recipients of Tg4.PD-1^−/−^ vs., Tg4.PD-1^+/+^ T cells?

3) Studies in Figure 4 show that PIT can induce tolerance in an EAE transfer model in which Tg4 effector T cells are used. There are marked reductions in the frequencies of splenic IFN-g and GM-CSF producing Tg4 cells. The manuscript would be strengthened by comparisons of the effects of PIT on naïve and Teff cells. Are there similar differences in frequency in IL-17 producing Tg4 cells? How do the total numbers of IFN-g, GM-CSF, and IL-17 producing Tg4 cells compare in PIT and PBS treated mice? Are there any changes in FoxP3^+^ frequencies and the Treg/Teff cell ratios in the PIT vs. PBS treated mice?

4) In Figure 5, the authors should clarify whether PD-1 expression is being examined on spleens after secondary transfer from recipients in which EAE was induced or in mice in which was not induced. The figure legend is not clear, and the text states that PD-1 expression is maintained in secondary hosts that had not been exposed to PIT, but lacks sufficient detail. Is the presence of antigen needed to maintain PD-1 expression in PIT treated effector Tg4 cells?

5) Studies in Figure 6 show that PIT treatment alters DNA methylation patterns at the *Pdcd1* promoter and reduces 5hmC enrichment at the *Pdcd1* promoter CR-B and CR-C regions. Further work tested whether changes in the relative expression of 5hmC at the *Pdcd1* promoter expression was reflected by alterations in expression of mammalian Ten-Eleven-Translocation proteins. It is not clear if any of the differences in TET gene expression (most are 2 fold changes) are significant from the data presented; however, the authors state that they have identified dynamic changes in TET gene expression.

6) Previous work from these authors indicated that PD-1 was dispensable for both induction and maintenance of tolerance in PIT exposed naïve CD4^+^ T cells. What leads to apoptosis in the published model, but persistence of T cells in the models presented in this manuscript?

---

## [Author Response]

After careful consideration of the comments provided, we are fully confident that we can revise the manuscript to address them, by providing additional data that we already have to-hand, and by undertaking some additional experiments to address reviewer 1’s request for data from a non-TCR transgenic model.

[Editors’ note: the authors provide an outline of their proposals before resubmission and than an update on the work completed.]

Reviewer #1:

*This report shows that peptide immunotherapy (PIT) of TCR-transgenic mice induces PD-1-expressing anergic T cells from naïve or effector CD4*^*+*^
*T cells and that DNA demethylation of the promoter region of the PD-1 gene is characteristic of such anergic T cells. This sustained PD-1 high expression accompanying epigenetic change in transgenic CD4*^*+*^
*T cells after PIT is interesting. Other findings in*
Figures 1, 2, 3 and 4
*are not so novel*.

Figures 1, 3 and 4 include necessary information describing the T cell transfer PIT model for naïve T cells and Teff cell respectively. They include data pertinent to the form of tolerance induced (e.g. the lack of Foxp3 expression in donor cells (Figure 1), the degree of donor cell persistence and inflammatory cytokine production, PD-1 expression and the resistance to tolerance seen by using Tg4.PD-1^−/−^ T cells (Figures 1, 3 and 4). Figure 2 highlights the maintenance of pMHC complexes specifically on CD4^+^ DC after PIT, which to our knowledge is entirely novel. Most importantly, the key novel aspect of this work is that we provide data on the mechanistic basis for PIT induced unresponsiveness in Teff cells, rather than the commonly used naïve T cells.

*Specific comments*:

*1) In*
Figure 5*, it is not clear how PIT induces high expression of PD-1 in CD4*^*+*^
*T cells*.

PD-1 is upregulated in response to TCR stimulation (1).

We contend in the manuscript, the stable expression of PD-1 is due to epigenetic changes in the *Pdcd1* promoter in response to the TCR stimulation provided by PIT.

*Is it primarily attributed to high expression of PD-L1 by DCs presenting peptide/MHC*?

We are not arguing that the PD-L1 expressed by DC is important for the sustained PD-1 expression by T cells. It is not clear why such a feedback mechanism would exist. During productive T cell immune responses, T cells transiently express PD-1 and activated DC upregulate PD-L1. Thus, if such a feedback mechanism were possible, sustained PD-1 expression by T cells might ensue during normal immunity, which is not seen and would clearly be undesirable. Rather, we propose that the constitutive expression of PD-L1 by CD4^+^ DC allows them to provide a dominant negative signal to the T cells, by ligating PD-1.

*If so, how does PIT induce high PD-L1 in DCs*?

We have no evidence that PIT induces high PD-L1 in DC.

Please note that the data shown in Figure 2 are from analyses of steady state DC (i.e. from mice that had not received PIT). This level of PDL-1 was not increased by administration of peptide using the PIT protocol (data not shown). Thus the key point is that CD4^+^ DC have constitutively high expression of PD-L1. This is in fact evident (but not considered/discussed) in other published reports (e.g. [56]).

We therefore conclude that the PIT effect is driven by soluble peptide being concentrated on these DC (as indicated in Figure 2). The mechanism for this concentration is unknown, but clearly this DC population merits further investigation as the reviewer suggests below.

*Can peptide/MHC-expressing PD-L1+ DCs induce anergy in transgenic CD4*^*+*^
*T cells in vitro? Does transfer of such peptide-pulsed PD-L1-high DCs induce PD-1-high anergic T cells in vivo*?

As you will appreciate, the CD4^+^ DC population is quite a rare fraction within the total splenic mononuclear population (Figure 2 of the manuscript). However, we agree that we should attempt to discern any inherent “tolerogenic” function of these cells after their isolation.

Suggested experiments:

It is unclear whether we can isolate sufficient numbers for in vivo transfers. We therefore propose to address whether these cells can influence Tg4 Teff function in vitro. If this is successful, we will endeavour to perform in vivo transfers. Though they will likely require large mouse numbers for DC donors, these experiments would not be long-term and could be completed in a reasonable timeframe.

We should point out that there is one important caveat that makes it quite unclear whether we will in fact see any “tolerogenic” effect using isolated CD4^+^ DC. This is that DC have the propensity to become activated during their isolation ex vivo, upregulating costimulatory molecules such as CD80, CD86 and CD40 (Schlecht et al., 2006; Vremec et al., 2011).

We cannot therefore be certain that the functional status of these DC in vitro will reflect that displayed in their steady state in vivo. Relevant to this, we used isolated DC populations to stimulate the proliferation of a Tg4 cell line (which are essentially Teff) to identify CD4^+^ DC as the APC fraction that holds peptide-MHC complexes after peptide administration (Figure 2), suggesting that these cells may have been partially activated during the purification process.

Therefore, although we can perform the additional experiments that we describe above, at this point we cannot predict how this well-established confounding effect of DC isolation will influence the results.

UPDATE: We performed a series of experiment assessing the effects of DC and macrophage populations on Tg4 Teff cells in vitro. In order to mimic in vivo timings used in the paper, we stimulated for 4 days with or without MBP peptide. As can be seen in Figure 7, all DC populations (CD4^+^CD8_-_, CD8^+^CD4^-^ and pDC) as well as macrophages were equally capable of activation Tg4 Teff as assessed by CD44 upregulation. All APC populations also led to strongly increased PD-1 expression. Most tellingly, CD4^+^ DC did not drive a loss in the ability of Tg4 Teff to produce proinflammatiory cytokines (IFN-γ and GM-SCF). Therefore these in vitro experiments using isolated DC were unable to recapitulate the tolerant nature of Tg4 Teff exposed to PIT in vivo. We conclude that, as we anticipated in our previous letter, activation of DC during their ex vivo isolation precludes any meaningful development of an in vitro model to determine/mimic their in vivo tolerogenic function. Based on this, and the comparatively rare nature of the CD4^+^ DC population, any follow-on attempts at using these cells in transfer models seem impractical. We have not included these data in the revised manuscript.Author response image 1.In vitro DC: T cell cultures. Tg4 Teff (CD45.1) were rested for one day prior to culture with FACS-sorted APC populations from the spleens of C57BL/6xB10.PL (CD45.2) mice in the presence of absence of 2µg/ml Ac1-9. Cells were analyzed four days later. B & C PD-1 and CD44 expression on Tg4 Teff cells. D & E effector cytokine production after overnight recall stimulation with 20µg/ml Ac1-9 and fresh irradiated C57BL/6xB10.PL splenocytes.

*2) Can these findings with transgenic T cells be extended to non-transgenic mice? For example, does PIT induce MBP-tetramer+ PD-1-high anergic T cells in normal mice*?

This is challenging because MBP tetramers are not available to us. However, there is an alternative approach.

Suggested experiments:

We propose to use the MOG-induced EAE model in C57BL/6 mice. Passive EAE can be transferred using T effector cells taken from lymph node cells from MOG-immunized donor mice (O'Connor et al., 2008). These can be transferred prior to PIT/vehicle administration and the donor cells assessed for PD-1 expression, as we have done for Tg4 cells in the original manuscript.

We shall use a congenic marker to distinguish the transferred cells (CD90.2 cells into CD90.1 hosts). Please note that we would not have the absolute purity that the Tg4 transfer system allows (i.e. not all of the transferred T cells will be MOG-responsive), so we do not propose to undertake downstream epigenetic studies, which would also be time-consuming. However, this plan should be sufficient to identify any PD-1hi population that emerges in donor cells following exposure to PIT versus PBS, as the reviewer suggests.

UPDATE: For these experiments we chose to slightly modify the proposed approach, by focusing on ovalbumin (323-339)-driven T cell responses. The reason for this was that we had access to an (untested) I-Ab/OVA(323-339) tetramer. Unfortunately, tetramer-staining was not reliable due apparent non-specific staining of host T cells. Despite this, we were able to gain clear evidence that PIT using OVA(323-339) had induced upregulation of PD-1 amongst transferred Teff cells (pOVA-primed CD45.1+ Teff cells into CD45.2 hosts). Moreover, intracellular staining showed that PIT clearly inhibited the ability of donor Teff cells to produce pro-inflammatory cytokines. These data have been included as Figure 5—figure supplement 7 and Figure 5—figure supplement 8, and are described in an extended Results section.

Reviewer #2:

*1) Studies in*
Figure 1
*show that administration of MBP peptide i.v. leads to persistence of Ag-specific donor T cells. Data show that IFN-g and IL-17 production was reduced in splenic recall assays. Is this due to a difference in cytokine production on a per cell basis or the frequency of the cytokine producing cells? How do the percentages (and numbers) of IFN-g and IL-17 producing cells in PIT-treated and control mice compare? Also is there a difference in the frequency of IL-10 producing cells? Is T cell survival affected by PIT in this model*?

As shown in Figure 1 (upper left panel) naïve Tg4 cell numbers increase in response to PIT, prior to immunization. Following immunization, Tg4 numbers in PIT- and PBS-treated mice become equivalent (Figure 1, upper right panel), reflecting the immunization-driven expansion that would be expected in the PBS-treated group, in the absence of prior exposure to PIT. This also means that there is no dramatic loss of PIT-exposed T cells following secondary exposure to TCR stimulation at immunization; numbers of Tg4 cells in the PIT-treated mice are similar before immunization (Figure 1 upper left) and after immunization (Figure 1 upper right).

Intracellular staining (ICS) revealed that the frequencies and numbers of IL-17+ donor Tg4 cells were significantly reduced following PIT and there was a similar trend for IFN-γ+ cells. We were unable to detect IL-10+ cells by ICS. IL-10 secretion, measured by ELISA, was low from PBS-exposed cells and undetectable from those exposed to PIT.

We conclude that PIT reduces the frequencies and numbers of donor cells producing inflammatory cytokines as well as the total amount secreted. This also seems to be the case for IL-10, with no evidence for elevation in its production in response to PIT.

Suggested addition:

The additional ICS cytokine data can be incorporated into a new Figure 1—figure supplement 1.

The IL-10 ELISA data can be included into Figure 1 of the manuscript, to complement the existing data on IFN-γ and IL-17.

UPDATE: The IL-10 ELISA data have been added to Figure 1 of the manuscript. The additional ICS cytokine data are now incorporated into a new Figure 1—figure supplement 1.

*2) Studies in*
Figure 3
*show that PIT treatment of recipients of naïve Tg4.PD-1*^*−/−*^
*T cells do not protect from EAE in contrast to PIT treatment of recipients of Tg4.PD*^*−1+/+*^
*T cells. This is a key finding that merits deeper analyses. How do Tg4T cell numbers, as well as percentages of FoxP3*^*+*^*, IFN-g and IL-17+ Tg4T cells, in the spleen and CNS compare in the PIT treated recipients of Tg4.PD-1*^*−/−*^
*vs., Tg4.PD*^*−1+/+*^
*T cells*?

Because of the therapeutic scenario that we are trying to model, we have focused our comparison upon PD-1-sufficient versus PD-1-deficient Tg4 Teff cells, rather than naïve T cells. There were no gross differences in the numbers or frequencies of donor cells in PBS-treated hosts, indicating that PD-1 expression does not constrain the accumulation of donor cells in the absence of PIT. However, when comparing donor cells in the spleens of PBS-treated versus PIT-treated hosts, the elevation seen in PIT-treated WT cells (shown in the manuscript Figure 4) was accentuated when donor cells were PD-1 deficient.

We saw no PIT-driven elevation in the frequencies of Foxp3^+^ cells amongst the donor cells, although PIT-exposed PD-1^−/−^ donor cells showed elevated numbers of Foxp3^+^ cells as a reflection of their elevated total numbers. The frequencies of proinflammatory cytokine+ cells (ICS) were reduced regardless of the PD-1-status of the donor Tg4 cells. However, again because of their increased total numbers, the numbers of cytokine+ donor cells were highest in PIT-treated hosts that had received PD-1^−/−^ donor cells.

Thus the inability of PIT to prevent EAE when the potentially pathogenic cells were unable to express PD-1 correlated with enhanced clonal expansion and therefore elevated total numbers of cytokine-capable T cells, rather than a failure of PIT to dampen the relative ability to produce cytokines.

We cannot accurately assess cytokine production in the CNS of PIT-treated mice that received WT Tg4 Teff, because these cells do not accumulate sufficiently within the CNS to provoke EAE (manuscript Figure 4).

Suggested addition:

We would add these data as a new Figure 5—figure supplement 1.

UPDATE: The data discussed are now incorporated as Figure 5—figure supplement 1, Figure 5—figure supplement 2 and Figure 5—figure supplement 3.

*3) Studies in*
Figure 4
*show that PIT can induce tolerance in an EAE transfer model in which Tg4 effector T cells are used. There are marked reductions in the frequencies of splenic IFN-g and GM-CSF producing Tg4 cells. The manuscript would be strengthened by comparisons of the effects of PIT on naïve and Teff cells. Are there similar differences in frequency in IL-17 producing Tg4 cells? How do the total numbers of IFN-g, GM-CSF, and IL-17 producing Tg4 cells compare in PIT and PBS treated mice? Are there any changes in FoxP3*^*+*^
*frequencies and the Treg/Teff cell ratios in the PIT vs. PBS treated mice*?

Frequencies and numbers of IFN-γ+ and GM-CSF+ donor cells can be incorporated into Figure 5—figure supplement 1, as described above. Please note that the in vitro polarization protocol used drives the encephalitogenic Tg4 Teff cells to produce IFN-γ and GM-CSF, but not IL-17 (O'Connor et al., 2008). The frequencies of Foxp3^+^ cells in the donor cell population when transferring naïve T cells are shown in the manuscript Figure 1, lower left panels.

There was also no elevation in Foxp3^+^ frequencies amongst transferred Teff cells in response to PIT.

Suggested addition:

We can incorporate these Foxp3 data into the manuscript, Figure 4.

With the addition of the data discussed above, the manuscript will allow the suggested comparison of the effects of PIT upon naïve T cells versus Teff cells. We can incorporate additional text to direct the reader to the appropriate figures. However, if the editors consider it of value, we could include further (repeat) data that are already in-hand to show naïve and Teff cells within the same figure. Please note that those data would not be from experiments performed at the same time, but the patterns seen with naïve and Teff cells are robust across a series of experiments. We would value the Editors’ advice as to whether they consider a direct comparison of the two cell types in the same experiment to be essential.

UPDATE: These Foxp3 data have been added to Figure 4.

*4) In*
Figure 5*, the authors should clarify whether PD-1 expression is being examined on spleens after secondary transfer from recipients in which EAE was induced or in mice in which was not induced. The figure legend is not clear, and the text states that PD-1 expression is maintained in secondary hosts that had not been exposed to PIT, but lacks sufficient detail. Is the presence of antigen needed to maintain PD-1 expression in PIT treated effector Tg4 cells*?

The experiments shown in Figure 5 were designed to address this very question (is antigen needed to maintain PD-1?). Therefore, donor Tg4 Teff cells were re-isolated from primary hosts that had received either PIT or PBS.

These cells were then administered to secondary hosts to test i) whether these cells could induce EAE (Figure 5) and ii) whether the PIT-exposed Teff cells would maintain PD-1 in the absence of peptide administration in the secondary host (i.e. these secondary hosts did not receive PIT). The results in Figure 5 show that, in PIT exposed Teff, PD-1 is indeed maintained in the absence of further exposure to antigen.

Suggested change:

We can re-word the figure legend and the text in the Results section to make these points explicitly clear.

UPDATE: We have re-worded the legend to Figure 5 and the text.

*5) Studies in*
Figure 6
*show that PIT treatment alters DNA methylation patterns at the* Pdcd1 *promoter and reduces 5hmC enrichment at the* Pdcd1 *promoter CR-B and CR-C regions. Further work tested whether changes in the relative expression of 5hmC at the* Pdcd1 *promoter expression was reflected by alterations in expression of mammalian Ten-Eleven-Translocation proteins. It is not clear if any of the differences in TET gene expression (most are 2 fold changes) are significant from the data presented; however, the authors state that they have identified dynamic changes in TET gene expression*.

The key point is that Tet mRNA expression was almost entirely extinguished at the Teff stage (particularly for Tet1 and Tet3), but then regained in the PBS and PIT groups (i.e. expression was dynamic) (Figure 6). Thus, in the model we propose in the manuscript, cells at the Teff stage have an impaired ability to actively demethylate cytosine, which is consistent with the data shown in Figure 6. Please note that the values displayed are from qPCR comparing expression of TET mRNA relative to that of the housekeeping gene.

Thus the drop in TET mRNA expression at the Teff stage is far greater than 2-fold; e.g. the change in Tet 1 expression from naïve (relative expression ∼3) to Teff (no amplification for Tet 1) is pronounced. Changes are statistically significant (p<0.05). This will be added to a revised Figure 6.

UPDATE: Significance bars have been added to Figure 6.

*6) Previous work from these authors indicated that PD-1 was dispensable for both induction and maintenance of tolerance in PIT exposed naïve CD4*^*+*^
*T cells. What leads to apoptosis in the published model, but persistence of T cells in the models presented in this manuscript*?

We do not currently have a definitive answer for this question. However, we would like to emphasize that this is rather tangential to the purpose of this study, which was to use Teff cells to better model the clinical scenario and to provide insights into the mechanistic basis for PIT in that setting. Tg4 cells serve as a well-established resource for analysis of PIT ([5]; Sundstedt et al., 2003; Anderson et al., 2006; Nicolson et al., 2006; Gabrysova et al., 2009), although the Tg4 cell transfer models that we describe here have not been utilized previously.

Hence, although we show the data on naïve Tg4 cells in Figures 1 and 3, these really serve as a prelude to the data using Teff cells. Using other models, a body of evidence is now indicating that Teff cells behave quite differently to naïve T cells in response to PIT. In particular, Teff appear capable of survival/expansion, rather than undergoing apoptosis (David et al., 2014; [32]). This is consistent with the expansion of Tg4 Teff that we report here.

The reviewer is correct to highlight that PD-1 was dispensable in our previous study of PIT in naïve T cells (24). Indeed, we discussed this difference in the manuscript, in the Discussion section. That model saw deletion of naïve T cells in response to PIT. Therefore, despite upregulation of PD-1 by PIT exposed T cells, there was no opportunity for this to mediate tolerance, as the cells died. This explains why PD-1 was not required in that model for the maintenance of tolerance (the cells had been deleted).

Of particular note, that previous study, used ovalbumin-responsive TCR transgenic T cells, OT-II.

We have recently used OT-II T cells that had been polarized towards Th2 Teff to study PIT in the setting of allergic airways inflammation (32). Consistent with the current study, there we also saw that Teff cells survived and expanded in number in response to PIT, emphasising the differences between naïve and Teff cells, using cells from the same TCR transgenic source.

Therefore, the survival of Tg4 Teff that we observe here is consistent with the developing paradigm (Campbell et al., 2009; David et al., 2014; [32]). We have previously discussed in detail the molecular basis for the apoptosis of naïve T cells in response to PIT and can insert a suitable review reference (Hochweller et al., 2006).

The reasons why naïve Tg4 cells are not deleted by PIT merit detailed interrogation. However, such analyses represent a long-term project, beyond the scope of the current manuscript, which has focused on mechanistic studies using Teff cells.

*Further new data*:

The editors requested that we addressed how PIT induces PD-1 dependent anergy. We have performed additional experiments comparing the response profiles of Tg4.PD^−1+/+^ versus Tg4.PD-1^−/−^ Teff to PIT. As can be seen in the new Figure 5—figure supplement 1, there is a marked expansion in the numbers of Tg4.PD-1^−/−^ Teff cells in response to PIT. This is indicative of unrestrained clonal expansion in the absence of PD-1 signalling. To explore this further, we have assessed the ability of the Tg4 Teff to express CD25, and to produce and respond to IL-2. Upon ex vivo retrieval (4 days after PIT or PBS) levels of CD25 expression were low, irrespective of treatment group. CD25 expression was upregulated by in vitro stimulation with MBP peptide. Notably, this CD25 upregulation was impaired in the PIT-treated Tg4.PD^−1+/+^ Teff cells, but was intact in PIT-treated Tg4.PD-1^−/−^ Teff cells. As a consequence phosphorylation of STAT5 upon exposure to IL-2 in PIT-treated Tg4.PD-1^−/−^ Teff cells was strong (at least equivalent to PBS-treated Teff cells), but was impaired in PIT-treated Tg4.PD^−1+/+^ Teff cells. These data suggest that a key function of PD-1 during PIT is to restrict the capacity of Teff to proliferate through, impaired CD25 upregulation and STAT5 signaling.

We have added these data as Figure 5—figure supplement 1, Figure 5—figure supplement 2, Figure 5—figure supplement 3, Figure 5—figure supplement 4, Figure 5—figure supplement 5 and Figure 5—figure supplement 6

We have added an additional section to the Results, describing these data and extended the Discussion.